# Higher gametocyte production and mosquito infectivity in chronic compared to incident *Plasmodium falciparum* infections

Aissata Barry[1,2,7], John Bradley [3,7], Will Stone[4,7], Moussa W. Guelbeogo[1], Kjerstin Lanke[2], Alphonse Ouedraogo[1], Issiaka Soulama [1], Issa Nébié[1], Samuel S. Serme[1], Lynn Grignard [4], Catriona Patterson [4], Lindsey Wu[4], Jessica J. Briggs [5], Owen Janson[5], Shehu S. Awandu[2], Mireille Ouedraogo[1], Casimire W. Tarama [1], Désiré Kargougou[1], Soumanaba Zongo[1], Sodiomon B. Sirima[1], Matthias Marti [6], Chris Drakeley [3], Alfred B. Tiono[1] & Teun Bousema[2✉]

*Plasmodium falciparum* gametocyte kinetics and infectivity may differ between chronic and incident infections. In the current study, we assess parasite kinetics and infectivity to mosquitoes among children (aged 5–10 years) from Burkina Faso with (a) incident infections following parasite clearance ($n = 48$) and (b) chronic asymptomatic infections ($n = 60$). In the incident infection cohort, 92% (44/48) of children develop symptoms within 35 days, compared to 23% (14/60) in the chronic cohort. All individuals with chronic infection carried gametocytes or developed them during follow-up, whereas only 35% (17/48) in the incident cohort produce gametocytes before becoming symptomatic and receiving treatment. Parasite multiplication rate (PMR) and the relative abundance *of ap2-g* and *gexp-5* transcripts are positively associated with gametocyte production. Antibody responses are higher and PMR lower in chronic infections. The presence of symptoms and sexual stage immune responses are associated with reductions in gametocyte infectivity to mosquitoes. We observe that most incident infections require treatment before the density of mature gametocytes is sufficient to infect mosquitoes. In contrast, chronic, asymptomatic infections represent a significant source of mosquito infections. Our observations support the notion that malaria transmission reduction may be expedited by enhanced case management, involving both symptom-screening and infection detection.

[1] Centre National de Recherche et de Formation sur le Paludisme (CNRFP), Ouagadougou 01, Burkina Faso. [2] Radboud Institute for Health Sciences and Radboud Center for Infectious Diseases, Radboud University Medical Centre, Nijmegen, The Netherlands. [3] MRC International Statistics and Epidemiology Group, London School of Hygiene and Tropical Medicine, London, UK. [4] Department of Immunology and Infection, London School of Hygiene and Tropical Medicine, London, UK. [5] Department of Medicine, University of California San Francisco, San Francisco, CA, USA. [6] Wellcome Centre for Integrative Parasitology, University of Glasgow, Glasgow, UK. [7] These authors contributed equally: Aissata Barry, John Bradley, Will Stone. ✉email: teun.bousema@radboudumc.nl

The epidemiology of *P. falciparum* transmission stages, gametocytes, is poorly understood. Molecular diagnostics show that low-density gametocyte carriage is highly prevalent in endemic populations[1] and that many low-density infections are infectious to mosquitoes[2], explaining observations from the 1950s that gametocyte-free individuals (as determined by microscopy) frequently infected mosquitoes[3]. Few assessments of the population infectious reservoir of malaria parasites have been undertaken[4,5], particularly with tools capable of detecting low parasite and gametocyte densities[6,7]. Fewer still have been able to assess the association of infectivity with factors other than parasite density, despite clear evidence that gametocyte maturity[8,9], intrinsic parasite factors[10,11], human genetic factors[12], and human clinical and immune responses[13–15] can have significant influence on the effectiveness of parasite transmission.

Higher densities of pathogenic asexual stage malaria parasites are commonly associated with the presentation of symptoms[16]. Because gametocytes develop from asexual parasites, individuals with clinical malaria and higher parasite densities may have more gametocytes and be more likely to infect mosquitoes[17,18]. On the other hand, the long maturation process of gametocytes may result in higher gametocyte densities in infections of longer duration that are often asymptomatic[6,19]. These uncertainties have added to the ongoing debate over the importance of asymptomatic infections for malaria transmission[19–21]. To truly understand the interaction of symptomatic status and the temporal dynamics of malaria infection, longitudinal assessments are required that can determine whether infected individuals living in malaria-endemic areas are infectious to mosquitoes before symptoms arise or before malaria infections are detectable with standard diagnostics. Such longitudinal studies can also prospectively assess how gametocyte production is influenced by infection characteristics such as parasite multiplication rates[22], clinical symptoms, duration of infection and complexity of infection[23].

In the present longitudinal study, we aimed to describe changes in asexual and sexual parasite density and infectiousness at two moments of the parasite's natural history: immediately after blood-stage infection establishment, and during the chronic phase of the infection. We hypothesized that a large proportion of incident infections would develop into chronic asymptomatic infections that are highly infectious to mosquitoes and that both host and parasite factors influence gametocyte production and infectivity. To test these hypotheses, we recruited two cohorts of school children from Balonghin, Burkina Faso. In the first cohort, aiming to characterize incident infection dynamics, individuals without infections were monitored weekly by PCR for up to 6 months to detect new infections at their onset. In the second cohort, aiming to characterize chronic asymptomatic infections, individuals were enroled for monthly monitoring, and were classified as having chronic and asymptomatic infection when parasites were detected on consecutive visits by PCR in the absence of symptoms. Blood samples were taken to measure total parasite density, complexity of infection, parasite multiplication rates over 48-h intervals, asexual and sexual stage anti-malarial antibodies, and male and female gametocyte densities. Prospective gametocyte production was estimated by determining the ratio of gametocytes present at day 14 of follow-up to the density of ring-stage parasites at enrolment. The abundance of *P. falciparum* Apetala2-G (*ap2-g*; PF3D7_1222600) and Gametocyte exported protein 5 (*gexp-5*; PF3D7_0936600) gene transcripts were assessed as proxies of sexual conversion. Infectivity to mosquitoes was determined upon detection of incident or chronic asymptomatic infections, and at multiple time points during follow-up.

## Results

**Parasite and gametocyte kinetics differ between incident and chronic infections**. A total of 253 individuals were screened for participation in the incident infection cohort; 80 were confirmed *Plasmodium* negative by nested PCR (nPCR) that was conducted immediately on-site, and followed up weekly to monitor for infection (Table 1). Fifty-two individuals became *Plasmodium* positive, prompting their full enrolment into the cohort. The median observation period from nPCR confirmation of parasite negativity to infection detection and enrolment (day 0) was 27 days (interquartile range [IQR] 20–41). Four individuals were subsequently dropped from this analysis after quantitative PCR (qPCR), performed upon study completion, showed that *Plasmodium* parasites were present at time points more than 2 weeks prior to their detection by nPCR, leaving 48 individuals in the incident infection cohort. For the chronic infection cohort, 228 individuals were screened and 60 asymptomatic, parasite positive individuals were enroled. At enrolment, all participants in this cohort had been parasite positive for approximately 1 month (median 28.5 days [IQR 28–38.5]). Screening and follow-up time points are shown for all individuals in Supplementary Fig. 1. Blood samples were taken from individuals with confirmed incident or chronic infection at enrolment (day 0), then daily for 1 week, and weekly until day 35 or the presentation of malaria symptoms (Fig. 1). Infectivity to mosquitoes was determined at day 0, 14 and 35, or at the final visit if this preceded day 35 due to the presentation of symptoms.

At day 0, total parasite densities by qPCR were similar for chronic and incident infections (Table 1). Average parasite density trajectories for individuals in both cohorts are presented in Fig. 2A (data for individual participants is shown in Supplementary Fig. 2). During follow-up, median peak parasite density was higher in incident infections ($p = 0.0001$). At day 0, gametocytes were detected by qRT-PCR in 6% (3/48) of individuals with incident infections and 97% (58/60) of individuals with chronic infections. Only 35% (17/48) of individuals in the incident cohort developed detectable gametocytaemia during follow-up compared to 100% (60/60) of chronic infections. In both cohorts, there was considerable variation in gametocyte densities between individuals (Fig. 3A, B), but the median gametocyte density among positives at any time point ($p < 0.0001$) and the median peak gametocyte density were both significantly higher in chronic infections ($p < 0.0001$). Combining both cohorts, the association of gametocyte densities at day 14 with total parasite densities at day 0 (Spearman rho 0.48 $p = 0.0001$) (Fig. 3C) was significantly stronger than between gametocytes and total parasites at day 14 (Spearman rho $-0.19$ $p = 0.1292$) (Supplementary Fig. 3B). The intensity of sampling in the first week of follow-up allowed for assessments of parasite multiplication rate (PMR). PMR was calculated as the change in total parasite density between measures at 48-h intervals, giving multiple PMR observations per individual (Fig. 2B). PMR values varied widely within and between individuals. Geometric mean PMR was higher in individuals with incident infection (6.70, 95% CI [4.76–9.42]) compared to those with chronic infection (1.08 [0.90, 1.28]) ($p < 0.0001$); age and sex had no significant effect on this association. There was no evidence that PMR changed over time in the first week of intensive follow-up (incident, $p = 0.423$; chronic, $p = 0.338$). In the same period, gametocyte densities within individuals remained stable; the standard deviation (SD) of 48 hourly changes in gametocyte density was significantly lower than the SD of PMR in the same individuals ($p < 0.0001$) (Supplementary Fig. 3A). Among chronic infections, gametocyte densities decreased in the first week of observation (mean change over 48-h intervals = 0.87 [95% CI 0.77–0.97]).

Gametocyte sex ratio (the percentage of total gametocytes that were male) was significantly higher in chronic infections (median

**Table 1 : Characteristics and parasite metrics of participants.**

| Cohort | Year | N | Female, n (%) | Age, median (range) | Hb genotype, n (%) | | | | | Asexual density at baseline, median (IQR) | Peak asexual density, median (IQR) | Symptoms before day 35, n (%) | COI, median (IQR) | Gametocyte positive at baseline, n (%) |
| --- | --- | --- | --- | --- | --- | --- | --- | --- | --- | --- | --- | --- | --- | --- |
| | | | | | AA | AC | AS | SC | CC | | | | | |
| Incident | 2015 | 31 | 10 (32%) | 7 (5,10) | 19 (61%) | 10 (32%) | 2 (6%) | - | - | 85.7 (1.6–432.5) | 10839 (2529–7116) | 30 (97%) | 3 (1–4) | 1 (3%) |
| Incident | 2017 | 17 | 8 (47%) | 8 (5,10) | 7 (47%) | 6 (40%) | 1 (7%) | 1 (7%) | - | 201.9 (28.1–587.6) | 14193 (5689–21089) | 14 (82%) | 2 (1–3) | 2 (12%) |
| Incident | Total | 48 | 18 (38%) | 7 (5,10) | 26 (57%) | 16 (35%) | 3 (7%) | 1 (2%) | - | 127.1 (2.7–561.7) | 11593 (3373–40345) | 44 (92%) | 3 (1–4) | 3 (6%) |
| Chronic | 2016 | 40 | 17 (43%) | 8 (6,10) | 25 (64%) | 9 (23%) | 4 (10%) | 1 (3%) | - | 337.6 (91.2–1300.4) | 2320 (1067–8678) | 10 (25%) | 6 (3–9) | 38 (95%) |
| Chronic | 2017 | 20 | 9 (45%) | 9 (6,10) | 10 (53%) | 5 (26%) | 3 (16%) | - | 1 (5%) | 172.6 (84.2–386.4) | 1421 (607–3644) | 4 (20%) | 6 (6–7.5) | 20 (100%) |
| Chronic | Total | 60 | 22 (42%) | 8 (6,10) | 35 (60%) | 14 (24%) | 7 (12%) | 1 (2%) | 1 (2%) | 295.2 (92.2–796.3) | 1987 (770–5599) | 14 (23%) | 6 (5–9) | 58 (97%) |

Asexual parasite density was measured by 18S ribosomal DNA based qPCR[77]. Symptoms were defined as fever (>37.5 °C or reported fever) or other clinical manifestations of malaria infection. Complexity of infection (COI: number of parasite clones) was determined by deep sequencing of AMA-1[65] at the start of follow-up. Gametocyte densities were measured by quantification of female (Pfs25) and male (PfMGET) gametocyte specific mRNA by qRT-PCR[60].

38.1%) than incident infections (20.4%) ($p = 0.027$) (Supplementary Fig. 4). There was no association between haemoglobin (Hb) level at day 0 and gametocyte sex ratio at day 14 for participants with incident infections ($p = 0.482$), chronic asymptomatic infections ($p = 0.620$), or chronic infections that became symptomatic ($p = 0.116$).

**Infectivity to mosquitoes is higher among chronic infections.** 212 mosquito feeding assays were conducted over the course of the studies, with 52 assays in the incident cohort and 160 assays in the chronic cohort. Details of mosquito infection experiments are presented in Supplementary Table 3. The lower number of assays performed on individuals with incident infection reflects the earlier symptom presentation in this cohort. As demonstrated previously, there was a positive association between gametocyte density and mosquito infection rate (Fig. 4A, Spearman rho 0.51, $p < 0.0001$)[24]. Given that gametocyte densities were variable between and stable within individuals it is unsurprising that there was heterogeneity in infectivity between individuals ($p < 0.0001$). After adjusting for gametocyte density, there was still strong evidence for heterogeneity in infectivity between individuals ($p < 0.0001$), indicating that factors other than gametocyte density are also relevant in explaining transmission potential of infected individuals.

2/34 individuals with incident infection were infectious to mosquitoes on at least one occasion, resulting in 0.1% (5/3403) of mosquitoes becoming infected. By comparison, 28/60 individuals with chronic infection were infectious to mosquitoes on at least one occasion, infecting 4.5% (554/12,405) of mosquitoes (Supplementary Table 3). Gametocyte densities were generally higher in chronic infections, but after adjustment for gametocyte density the risk of mosquito infection for individuals in the chronic cohort remained higher (risk ratio = 10.44 [95% CI 2.79–39.01], $p = 0.0005$); this was true even after further adjusting for the presence of symptoms (risk ratio = 11.17 [95% CI 3.08–40.48], $p = 0.0002$). During chronic infections, there was no significant effect of assay timing (days after enrolment) on infectivity, whether adjusted ($p = 0.332$) or unadjusted ($p = 0.058$) for gametocyte density (Fig. 4B). Too few incident infections gave rise to mosquito infection to allow the same analysis; the two infectious feeds in this cohort were on days 1 and 16 after enrolment. There was no statistically significant correlation between total parasite density and infectivity (Spearman rho 0.12, $p = 0.081$), reflecting the observed lack of correlation between asexual and gametocyte densities. 88% of infectious individuals had total parasite densities above 50/μL (between the detection thresholds of expert field microscopy and standard RDTs[25]) (Supplementary Fig. 5). All infectious individuals had parasites above the estimated detection threshold of standard qualitative PCR (1 parasite/μL)[26]. There was no evidence that optimal sex ratios (between 10 and 50% male) were associated with higher mosquito infection rates ($p = 0.634$); the relationship between sex ratio and infectivity is complex, and has been explored in greater detail elsewhere[24].

**Complexity of infection is higher in chronic infections and associated with mosquito infection rates.** Targeted deep sequencing of apical membrane antigen 1 (AMA1) demonstrated that the majority of cohort participants had clonally complex infections at the start of intensive follow-up, both for incident infections (67% [31/46] multi-clonal, median complexity of infection (COI) = 3 [IQR 1–4]) and chronic infections (93% [51/55] multi-clonal, median COI = 6 [IQR 5–9]) (Table 1). The prevalence of multi-clonality ($p = 0.001$) and COI ($p < 0.0001$) was higher in chronic infections than incident infections at the

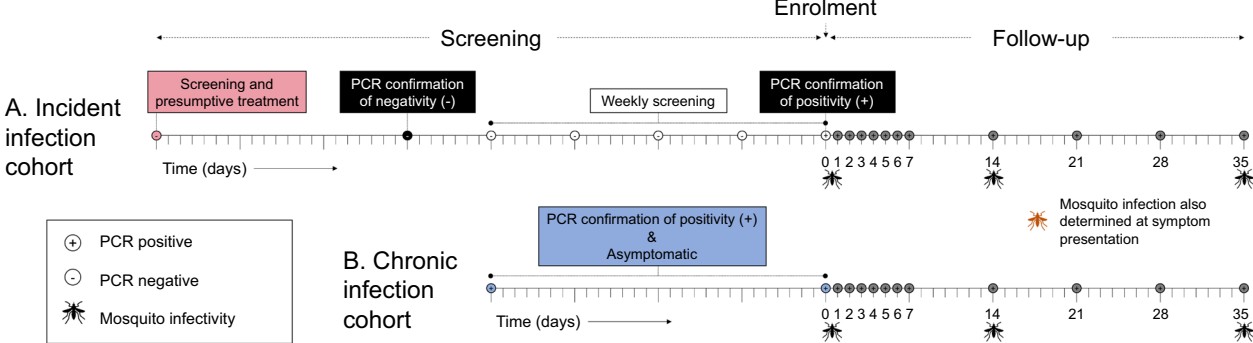

**Fig. 1 Illustrative sampling schemes for individuals in the incident and chronic cohorts.** Tick marks represent days and weeks, and sampling/screening time points are shown as circles. Positive (+) and negative (−) symbols inside circles indicate parasite status, as determined by the indicated assays. **A** In the incident infection cohort individuals were screened to confirm parasite negativity by microscopy, presumptively treated for sub-patent infection (light pink circle), and screened 3 weeks later with qualitative nested polymerase chain reaction-based on detection of *Plasmodium* specific 18s ribosomal DNA (nPCR) (black circle) to confirm the absence of sub-patent infection. They were then screened weekly with nPCR (white circles) until the onset of an infection. Intensive follow-up (grey circles) proceeded every day for 1 week, and weekly until day 35 after parasite detection. Study participants were closely monitored for the development of malaria symptoms. Participants were treated with artemether-lumefantrine upon the detection of symptoms or 35 days after initial detection of infection: whichever came first. **B** Individuals in the chronic infection cohort were screened monthly with nPCR to confirm chronic infection (blue circles), defined as two sequential visits with confirmed parasitaemia without any symptoms of malaria disease. Intensive follow-up proceeding from confirmation of chronic infection (grey circles) was as for incident infections. Participants were treated with artemether-lumefantrine upon the detection of symptoms or at day 35 of follow-up: whichever came first. Mosquito feeding (indicated with mosquito symbols) was conducted in both cohorts at the onset of intensive sampling (day 0) and at day 14 and 35 in the absence of symptoms. Mosquito feeding assays were additionally conducted on the day of symptom detection.

start of follow-up. Among the few incident infections that remained asymptomatic, 33% (2/6) and 100% (2/2) were multi-clonal at days 14 and 35, respectively. Chronic infections showed similar complexity throughout follow-up (day 14 = 98% multi-clonal [46/47], median COI = 6 [IQR 4–8], day 35 = 95% multi-clonal [39/41], median COI = 6 [IQR 4–8]). Neither complexity of infection at the start of follow-up ($p = 0.139$) nor Hb genotype ($p = 0.914$) were associated with PMR in the first week of follow-up. After controlling for gametocyte density and cohort, compared to those with 3 or fewer clones, those with 4–6 clones at the day of mosquito feeding were 4.23 [95% CI 1.37–13.10] times more likely to infect a mosquito, and those with 7 or more clones were 4.30 [95% CI 1.73–10.71] more likely ($p = 0.007$).

**Symptoms develop rapidly during incident infection and are associated with reduced infectivity to mosquitoes.** Most individuals with incident infections experienced symptoms prior to the end of follow-up at day 35 (92%, 44/48), compared to only 23% (14/60) of those with chronic infections. For those who developed symptoms, the median interval between day 0 and the onset of symptoms was 4 days (IQR 2–7) for incident infections, and 11 days (IQR 6–22) for chronic infections. At symptom presentation, median total parasite densities were 14,382/μL (IQR 2963–45,601) in chronic infections and 10,545/μL (IQR 1077–17,950) in incident infections ($p = 0.251$). Median gametocyte density at symptom presentation was 0/μL (IQR 0–0) in incident infections, and 5.3/μL (0.5–23.5) for chronic infections ($p = <0.0001$). In chronic infections that became symptomatic, total parasite densities increased during follow-up (average increase 6.0% per day [CI 0.08–15.2, $p = 0.023$, $n = 14$]), whereas for those that remained asymptomatic total parasite density decreased (average decrease 1.7% per day [CI 0.6–2.7%, $p = 0.002$, $n = 46$; $p$ value for difference <0.0001]). In incident infections that became symptomatic, total parasite densities increased during follow-up (average increase 73.0% per day [CI 58.3–88.0, $p = <0.0001$, $n = 44$]), whereas for those that remained asymptomatic total parasite densities remained level (0.2%

decrease per day [CI −6.3–6.3%, $p = 0.964$, $n = 4$; $p$ value for difference <0.0001]). PMR was higher in incident infections leading to symptoms than incident infections that remained asymptomatic ($p = 0.001$) or chronic infections with ($p < 0.001$) or without symptoms ($p < 0.001$). There was no evidence that PMR was different between incident infections that remained asymptomatic, chronic infections that remained asymptomatic, and chronic infections that became symptomatic ($p = 0.301$).

When mosquito feeding assays were conducted concurrently with the presentation of symptoms (25/212) infectivity was significantly decreased (risk ratio = 0.30 [95% CI 0.13–0.71], $p = 0.0061$); this was also the case after incident infections were excluded from the analysis (risk ratio = 0.26 [95% CI 0.11–0.65], $p = 0.004$).

**Factors associated with gametocyte production.** PMR and parasite density at enrolment (day 0) were independently positively associated with day 14 gametocyte density, whilst gametocyte densities were lower among individuals with incident infection and among older individuals (Table 2). The ratio of gametocyte density on day 14 to ring-stage parasite density on day 0 (i.e. the proportion of baseline asexual parasites that subsequently produced gametocytes) was significantly associated with the ratio of *ap2-g* transcript copy number to ring-stage parasite density on day 0 (i.e. the relative abundance of *ap2-g*) (Spearman rho 0.42, $p = 0.0015$) (Fig. 5A). This association was not observed when gametocyte production at day 14 was examined in relation to *ap2-g* production 6–8 days prior (Spearman rho = 0.04, $p = 0.780$), an interval that is too short to allow gametocyte production[27]. Relative *ap2-g* abundance on day 0 was significantly higher in chronic infections than in incident infections ($p < 0.0001$) (Fig. 5C). An independent marker of early sexual stage development, *gexp-5*, which is detectable in the cell cytoplasm within 14 h of gametocyte formation[28], was strongly associated with gametocyte production (Spearman correlation coefficient 0.70, $p = <0.0001$) (Fig. 5B), and was significantly higher in chronic infections ($p = <0.0001$) (Fig. 5D). Relative

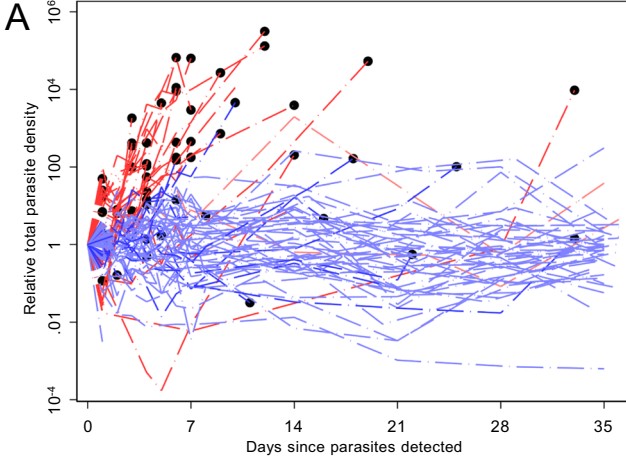

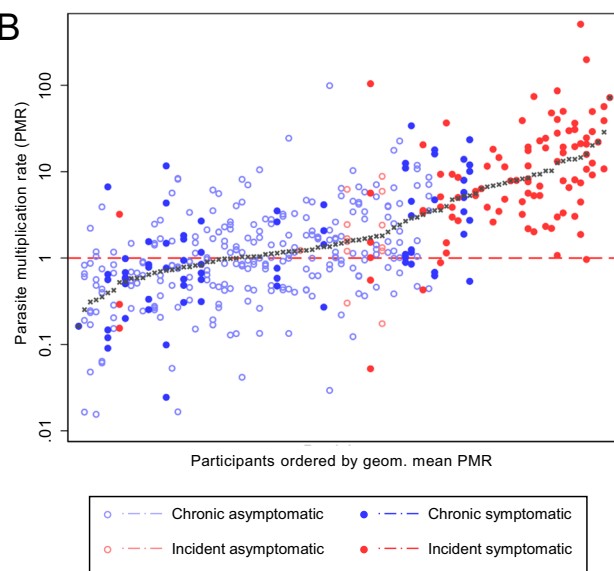

**Fig. 2 Infection trajectory and total parasite multiplication rate. A** Total parasite densities are presented as values relative to the time point of their first detection. Light blue lines = chronic asymptomatic infection, dark blue lines = chronic symptomatic infection, light red lines = incident asymptomatic infection, dark red lines = incident symptomatic infection, black markers = time of symptom onset. Three individuals became symptomatic at time points without concurrent parasite density measures. For these individuals, no black marker was drawn for 'Day of symptoms'. Full details of individual parasite and symptom trajectories are in Supplementary Figs. 1 and 2. **B** Total parasite multiplication rates (PMR) were calculated as the change in density between observations over 48-h intervals in the first 7 days of intensive follow-up. Marker colours are as for the lines in (**A**), with hollow circles for asymptomatic infection, solid circles for symptomatic infection, and grey crosses indicating the median PMR for each individual. The red dashed line at PMR = 1 indicates PMR equality between measures; values above show PMR values which increased over time, values below the PMR values which decreased over time. Participants are ordered by the geometric mean of their PMR observations. 431 measurements of PMR from 92 participants were available (median per person = 6 [IQR 3,6]). Incident asymptomatic: 13 observations (3 individuals), median observations = 6, IQR (1,6); Incident symptomatic: 98 observations (31 individuals), median observations = 3, IQR (2,4); Chronic asymptomatic: 247 observations (44 individuals), median observations = 6, IQR (6,6); chronic symptomatic: 73 observations (14 individuals), median observations = 6, IQR (5,6). Geom. mean = geometric mean.

*ap2-g* and *gexp-5* transcript abundance were correlated (Spearman rho 0.55, *p* = <0.0001). The associations of *ap2-g* and *gexp-5* transcript abundance with gametocyte production remained highly significant after adjustment for gametocyte density at the time of measurement (*p* < 0.0001).

**Antibody responses in relation to symptoms, parasite kinetics and mosquito infection rates.** The magnitude of antibody responses specific to 14 *P. falciparum* antigens was determined at enrolment for incident infections (*n* = 48) and chronic infections (*n* = 51). Antibody levels were higher in samples collected from individuals with chronic infections (*p* < 0.001 for all 14 antigens). Baseline antibody levels were significantly higher in individuals who remained asymptomatic for MSP2-ch150 (*p* = 0.036) in incident infections, and EBA-181 (*p* = 0.042) and GLURP-R2 (*p* = 0.032) in chronic infections (Fig. 6, Supplementary Table 1). Time to symptoms was not significantly longer in individuals with high antibody responses to pre-erythrocytic and asexual stage antigens in either the chronic cohort (*p* = 0.180) or the incident cohort (*p* = 0.203). In incident infections, antibody levels were negatively associated with PMR for EBA140 (*p* < 0.001), EBA-181 (*p* = 0.037), and PfAMA-1 (*p* = 0.030). Among individuals with chronic infections, antibody levels had no apparent association with mean PMR (Supplementary Table 2). In the chronic cohort infectivity was 65% (95% CI 21–85) lower for a 10-fold increase in anti-Pfs48/45 median fluorescence intensity (MFI) after adjustment for gametocyte density (*p* = 0.011); further adjusting for asexual immune responses infectivity was 48% (95% CI 5–81) lower for a 10-fold increase in anti-Pfs48/45 antibody response (*p* = 0.033). For Pfs230, infectivity was 64% (95% CI 34–80) lower for a 10-fold increase in MFI with adjustment for gametocyte density (*p* = 0.001), but this reduction was not significant after adjustment for asexual immune responses.

## Discussion
To maximize the impact of malaria control and elimination programmes, there is a need to identify which individuals are most important for transmission of *Plasmodium* to mosquitoes. Among our study population of school-aged children with intensively monitored incident and chronic *P. falciparum* infections, we observed marked variation in gametocyte production and mosquito infectivity. Incident infections were characterized by high parasite multiplication rates and low gametocyte commitment. More than 90% of incident infections resulted in a detectable fever within 35 days, which in most cases resulted in treatment before sufficient mature gametocytes were produced to infect mosquitoes. Gametocyte densities were significantly lower in incident infections, but even after adjustment for gametocyte density, the likelihood of transmission remained lower for incident infections and was further reduced by the presence of symptoms.

In the current study, children with incident and chronic infections were examined in detail for parasite kinetics and infectivity to mosquitoes. The study was conducted in an area of intense malaria transmission, confirmed by the observation that 65% (52/80) of parasite-free individuals became nPCR positive within 4 weeks (median participation 27 days [IQR 20–41]). These incident infections were frequently clonally complex at the time the infection was first detected in the bloodstream, indicating either co-transmission of multiple *P. falciparum* strains during mosquito bites[29,30] or repeated inoculations over a short time-period.

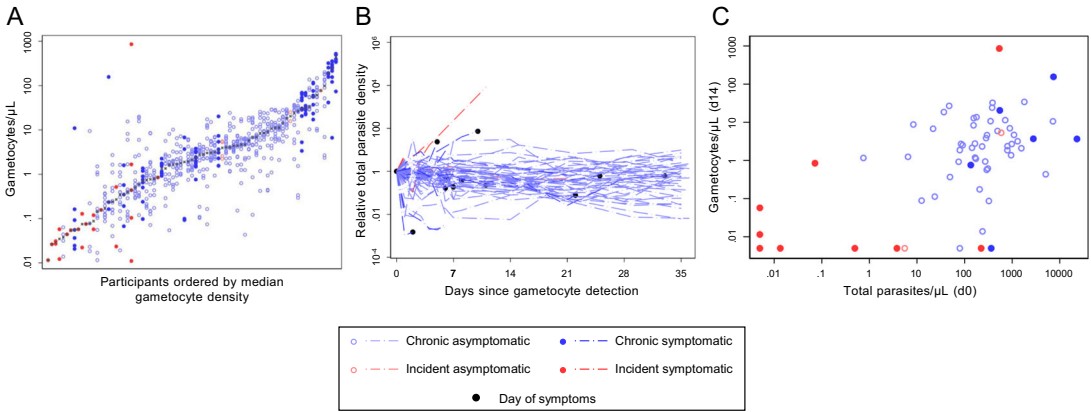

**Fig. 3 Gametocytes and gametocytogenesis. A** Gametocyte density measures from all gametocyte positive individuals (positive at any time point during follow-up). Light blue hollow markers = chronic asymptomatic infection, dark blue solid markers = chronic symptomatic infection, light red hollow markers = incident asymptomatic infection, dark red solid markers = incident symptomatic infection, grey crosses indicate the median gametocyte density for each individual. There were 928 measurements of gametocyte density; the median number of observations per person was 10 (IQR 5–12). For incident infections the median was 5 observations per person (IQR 3–8.5); for chronic infections the median was 12 (IQR 11–12). Individuals are ranked by median gametocyte density. **B** The association of gametocyte density at day 14 and total parasite density at enrolment (day 0). Time points are given as the number of days after enrolment into the incident or chronic infection cohorts. Line colours are as for the markers in (**A**), with black markers showing the onset of symptoms. Three individuals became symptomatic at time points without concurrent parasite density measures. For these individuals, no black marker was drawn for 'Day of symptoms'. Full details of individual parasite and symptom trajectories are in Supplementary Figs. 1 and 2. **C** Total parasite density at enrolment (d0) was positively associated with gametocyte densities at day 14 (Spearman rho 0.48, $p = 0.0001$). Colours are as in (**A**).

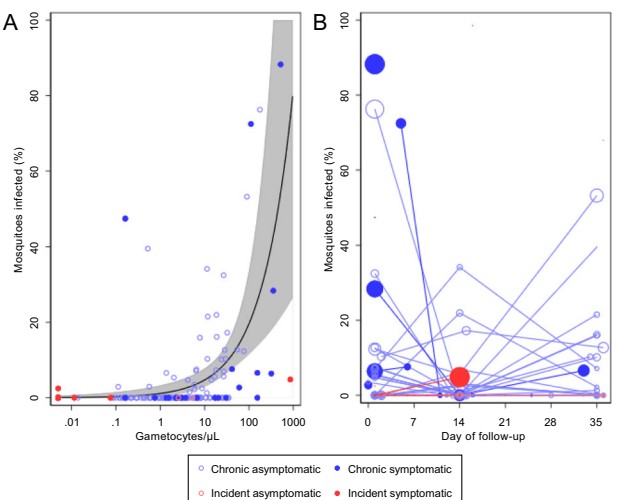

**Fig. 4 Mosquito infectivity and gametocyte density. A** Total gametocyte density in the different cohort participants in relation to the percentage of mosquitoes that became infected. Light blue hollow markers = chronic asymptomatic infection, dark blue solid markers = chronic symptomatic infection, light red hollow markers = incident asymptomatic infection, dark red solid markers = incident symptomatic infection. The black line indicates the shape of the model for the association of gametocyte density and percentage of mosquitoes infected in mosquito feeding assays for individuals with chronic infections. This model is described in detail elsewhere[24]. Grey shading represents the 95% confidence interval for the model. This figure is based on 134 observations from chronic asymptomatic infections, 26 observations from chronic symptomatic infections, 3 observations from incident asymptomatic infections, and 49 observations from incident symptomatic infections. **B** The percentage of mosquitoes infected over time, shown as days since the start of intensive follow-up. Colours (markers and lines) are as in (**A**). The size of the data points is proportional to the log gametocyte density in blood samples taken concurrent with the mosquito feeding assay. Connecting lines indicate observations from the same cohort participant.

The rapid infection rates in our study argue against the existence of sterile immune protection[31,32]. However, the high proportion of individuals with chronic infections who remained asymptomatic indicates that there was substantial clinical immunity in this age-group[33]. Antibody responses to specific blood-stage malaria antigens including EBA[34], GLURP[35] and MSP2[36,37] were associated with the ability to control parasite multiplication and remain symptom-free. In our incident infection cohort, the majority of individuals failed to control parasite multiplication and developed malaria symptoms (i.e. fever). Most incident infections occurred within 2 months (IQR 20–41 days) after presumptive treatment with dihydroartemisinin-piperaquine, when residual piperaquine levels are unlikely to provide effective prophylaxis[38]; piperaquine has no known effect on (developing) gametocytes[27,39]. In Mali, baseline parasitaemia did not influence the risk of developing clinical malaria[40]. It nevertheless remains to be confirmed whether the likelihood of detecting incident infections by repeated fever screening, that was high in the current study, is influenced by clearance of infections before the transmission season. Despite relatively small sample size, contrary to our expectations, haemoglobin genotype was not associated with parasite multiplication rates or the likelihood of remaining free of malaria symptoms upon infection[41,42]. This may have been due to our conservative definition of symptomatic malaria: the protection associated with HbS increases with the severity of disease[43], while in our study individuals were actively followed and treated immediately upon development of fever. Our finding that HbS did not affect PMR is in line with a controlled human infection study in Gabon where only the likelihood of symptoms and not PMR was lower for HbS individuals compared to those with normal haemoglobin[44].

Sexual conversion rates, quantified by relative *ap2-g* and *gexp-5* abundance[45], were higher in chronic infections than incident infections. This suggests that although gametocyte production may start during the first wave of erythrocytic schizogony[27,46], investment in asexual parasite multiplication dominates the early phase of infection while chronic infections are marked by higher investment in gametocyte production. This may be due to chronic inflammation and hence reduced levels of

**Table 2 Predictors of gametocyte density on day 14.**

| Covariate | Difference in log gametocytes/µl on day 14[a] | Adjusted difference in log gametocytes/µl on day 14[b] |
|---|---|---|
| Incident infection | −4.07 [−5.96, −2.18], p < 0.001 | −2.52 [−4.55, −0.50], p = 0.016 |
| Female | −0.95 [−2.30, 0.40], p = 0.163 | – |
| Age | | |
| 5–7 | 0 | 0 |
| 8–10 | −1.75 [−3.30, −0.19], p = 0.028 | −2.11 [−3.44, −0.77], p = 0.002 |
| Log total parasite density/µl on day 0 (1 unit increase) | 0.38 [0.13, 0.63], p = 0.004 | 0.52 [0.27, 0.77], p < 0.001 |
| Log total parasite density/µl on day 14 (1 unit increase) | 0.10 [−0.14, 0.35], p = 0.411 | – |
| Log PMR (1 unit increase) | 1.06 [0.05, 2.07], p = 0.040 | 1.57 [0.08, 3.07], p = 0.040 |
| Hb g/dl on day 14 (1 unit increase) | 0.29 [−0.22, 0.81], p = 0.256 | |
| Hb genotype | p = 0.931 | |
| AA | 0 | – |
| C | 0.31 [−1.36, 1.97] | – |
| S | 0.67 [−1.40, 2.73] | – |
| SC | 0.52 [−4.85, 5.89] | – |
| COI on day 0 | p = 0.239 | |
| 1–3 | 0 | – |
| 4–6 | −1.64 [−3.6, −0.40] | – |
| 7–19 | −0.86 [−3.01, −1.30] | – |
| Aggregate magnitude of antibody response | p = 0.285 | |
| Lowest tertile | 0 | – |
| Middle tertile | 0.40 [−1.33, 2.14] | – |
| Highest tertile | 2.27 [−0.57, 5.11] | – |

Data on day 14 gametocyte density was available for 65 individuals, 52 with chronic infections and 13 with incident infections. Hb genotype: haemoglobin genotype; COI: complexity of infection (number of parasite clones) on day 0. Aggregate magnitude of antibody response was calculated by ranking each individual according to their combined response to 13 *P. falciparum* antigens, as described in the 'Methods' section.
[a]Adjusted for whether the infection was incident.
[b]Adjusted for whether the infection was incident, log total parasite density/µl on day 0, log total parasite multiplication rate (PMR), and age.

Lysophosphatidylcholine (LysoPC) continuously triggering elevated rates of gametocyte production[47]. There is currently no in vivo data correlating LysoPC levels to inflammation and gametocyte production. Following future assessments of the contribution of systemic versus hematopoietic infection to gametocyte production, the implications of systematic inflammation to gametocyte commitment should be examined. Our observation that after adjusting for gametocyte density, the risk of mosquito infection was lower in incident infections suggests that gametocytes arising in the incident cohort were less mature at the time of mosquito feeding. The requirement for gametocyte maturation following release into the circulation would be in line with earlier in vitro observations that morphologically mature gametocytes may require several days of maturation to reach peak infectivity[48], and with findings in controlled human infections where infectivity is only observed several days after gametocyte densities plateau[39].

Importantly, many children in the incident infection cohorts developed symptoms of clinical malaria early upon infection. When symptoms were detected only 5% (2/42) of individuals had detectable gametocytes, with densities of 0.06/µL and 0.85/µL. Among chronic infections, marked variation in gametocyte production was observed that was positively associated with preceding total parasite density and multiplication rates, and negatively associated with age. The lower gametocyte production among older children was not explained by antibody responses to blood-stage antigens, potentially suggesting a role for recently described immune responses that reduce gametocyte maturation and increase with age[49]. Antibody responses to gametocyte/gamete antigens Pfs48/45 and Pfs230 were associated with reduced infectivity of successfully produced gametocytes[50]. Interestingly, infections with higher clonal complexity resulted in higher mosquito infection rates, independent of gametocyte densities. Although we observed a positive association between mosquito infection rates and COI before in the dry season in Burkina Faso[51], other studies found no association or a higher likelihood of mosquito infection for monoclonal infections[51,52]. The low gametocyte production among incident infections and the rapid abrogation of infection due to early treatment resulted in marked differences in the likelihood of mosquitoes becoming infected; only 0.1% of mosquitoes became infected in incident malaria infections compared to 4.5% in chronic malaria infections. Whilst this is largely explained by the number of gametocytes present in the blood at the time of feeding, we also observed that mosquito infection rates were lower when gametocytes arose early upon incident infections and when gametocyte donors had malaria symptoms[53]. The latter is in line with earlier findings, principally from animal models, on gametocyte-inactivating activity of inflammatory cytokines and reactive intermediates[54,55]. The mechanism by which this might be achieved and the relative importance for malaria transmission epidemiology requires further study.

From a public health perspective, our findings indicate that there is a window of opportunity to prevent onward malaria transmission from incident infections in semi-immune populations. The abundance, poor detectability, and high infectiousness of asymptomatic malaria infections suggest that these may be a considerable stumbling block for malaria elimination. However, if most infections are initially symptomatic (i.e. if the chronic infections we observed reflect the tail-end of symptomatic incident infections) enhancing case management to maximize accessibility of diagnosis and care may abrogate infections early on and, potentially, before individuals become infectious[56]. This is additionally supported by our observations that incident infections have initially low gametocyte production, are detectable by rigorous symptom-screening, and that gametocytes are less likely to achieve mosquito infection when arising early upon infection or when sampled during a clinical episode. Enhanced case management may thus reduce the proportion of infections that proceed to become highly infectious[57], in addition to the obvious clinical benefits of preventing progression to severe disease.

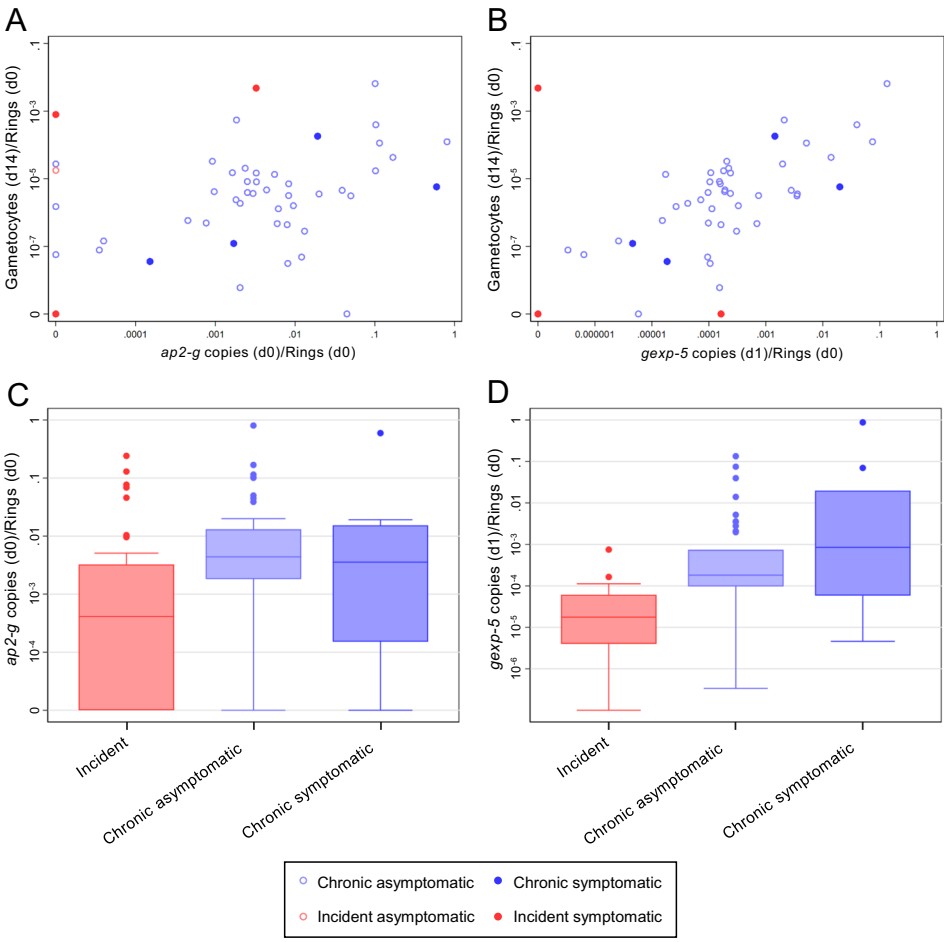

**Fig. 5 Sexual commitment and gametocyte production. A** Gametocyte production (y-axis) is given as the ratio of gametocytes/µL on day 14 to ring-stage parasites/µL (from *sbp1* qRT-PCR) on day 0. Sexual commitment (x-axis) was measured as the ratio of *ap2-g* mRNA copies to the density of ring-stage parasites/µL at day 0. Light blue hollow markers = chronic asymptomatic infection, dark blue solid markers = chronic symptomatic infection, light red hollow markers = incident asymptomatic infection, dark red solid markers = incident symptomatic infection. **B** Gametocyte production and relative abundance of *gexp-5* transcripts (ratio of *gexp-5* copy number to the density of ring-stage parasites/µL at day 0). Colours are as in (**A**). **C** Relative *ap2-g* abundance for the different cohorts; observations from incident symptomatic and incident asymptomatic infections were combined. This figure is thus based on 34 observations from combined incident infections, 41 observations from chronic asymptomatic infections, and 10 observations from chronic symptomatic infections. Box plot boxes span the median, 25th and 75th percentiles; box plot whiskers span the adjacent values with outliers shown as dots. **D** Relative *gexp-5* abundance and cohort; observations incident symptomatic and incident asymptomatic infections were combined. This figure is thus based on 22 observations from combined incident infections, 41 observations from chronic asymptomatic infections, and 10 observations from chronic symptomatic infections. Box plot boxes span the median, 25th and 75th percentiles; box plot whiskers span the adjacent values with outliers shown as dots.

## Methods

**Study location and screening**. The study protocol was approved by the ethical review boards of the London School of Hygiene and Tropical medicine (LSHTM) (#9008), the Centre National de Recherche et de Formation sur le Paludisme (CNRFP) (Deliberation number 2015-3-033) and the Burkina Faso National Ethical Committee for Health Research.

The study was conducted between June 2015 and December 2017 in Balonghin, in the health district of Saponé 45 kilometres Southwest of Ouagadougou, Burkina Faso. Balonghin experiences intense and highly seasonal malaria transmission from June to October. Children aged ≥5–10 years were screened for enrolment in the incident infection cohort in 2015 and 2017, and the chronic infection cohort in 2016 and 2017. 7/108 individuals participated in the study in two years, with the remainder participating in a single year. The purpose of the study and practical consequences of participation were explained during community meetings; screening was subsequently conducted at local schools. Individuals whose parent or guardian provided consent for screening were examined by a GCP-trained clinician. General criteria for enrolment in either cohort were that individuals were aged 5–10 years, were willing to provide repeated blood samples over a 6-month period, and if their caregivers provided informed consent. Exclusion criteria were complicated symptomatic malaria (defined according to standard World Health Organization criteria), anaemia (Hb < 8 g/dL), presence of any (chronic) illness that required immediate clinical care, family history of sudden death or of congenital or clinical conditions known to prolong QTc interval (e.g. family history of

symptomatic cardiac arrhythmias, clinically relevant bradycardia or severe cardiac disease), current treatment with drugs which could induce a lengthening of QT interval, known history of hypersensitivity, allergic or adverse reactions to piperaquine or other aminoquinolones, severe malnutrition (weight-for-height being below −3 standard deviation or <70% of median of the NCHS/WHO normalized reference values), weight below 15 kg, and current or previous participation in malaria vaccine trials.

**Sampling: Incident infection cohort**. During screening for the incident infection cohort, a blood smear was taken for malaria diagnosis. Individuals were eligible for enrolment if microscopy negative and were treated presumptively with dihydroartemisinin-piperaquine (Duocotexin®, Beijing Holley-Cotec Pharmaceutical, China, 40 mg dihydroartemisinin and 320 mg piperaquine tetra-phosphate per tablet) to clear possible sub-patent infections. Following confirmation of malaria parasite negativity 3 weeks later by *P. falciparum* 18S nested PCR (nPCR), individuals were visited weekly for clinical examination and infection status assessment by HRP2 and pLDH based RDT (RDT, First Response®, Premier Medical Corporation Ltd., Kachigam, India) microscopy and nPCR. This repeated sampling continued for up to 6 months (a total of ~25 sampling time points). Upon detection of an incident infection by nPCR, sampling was intensified to capture the dynamics of the infection shortly after the first appearance of parasites in the bloodstream and to monitor the development of high malaria parasitaemia and/or

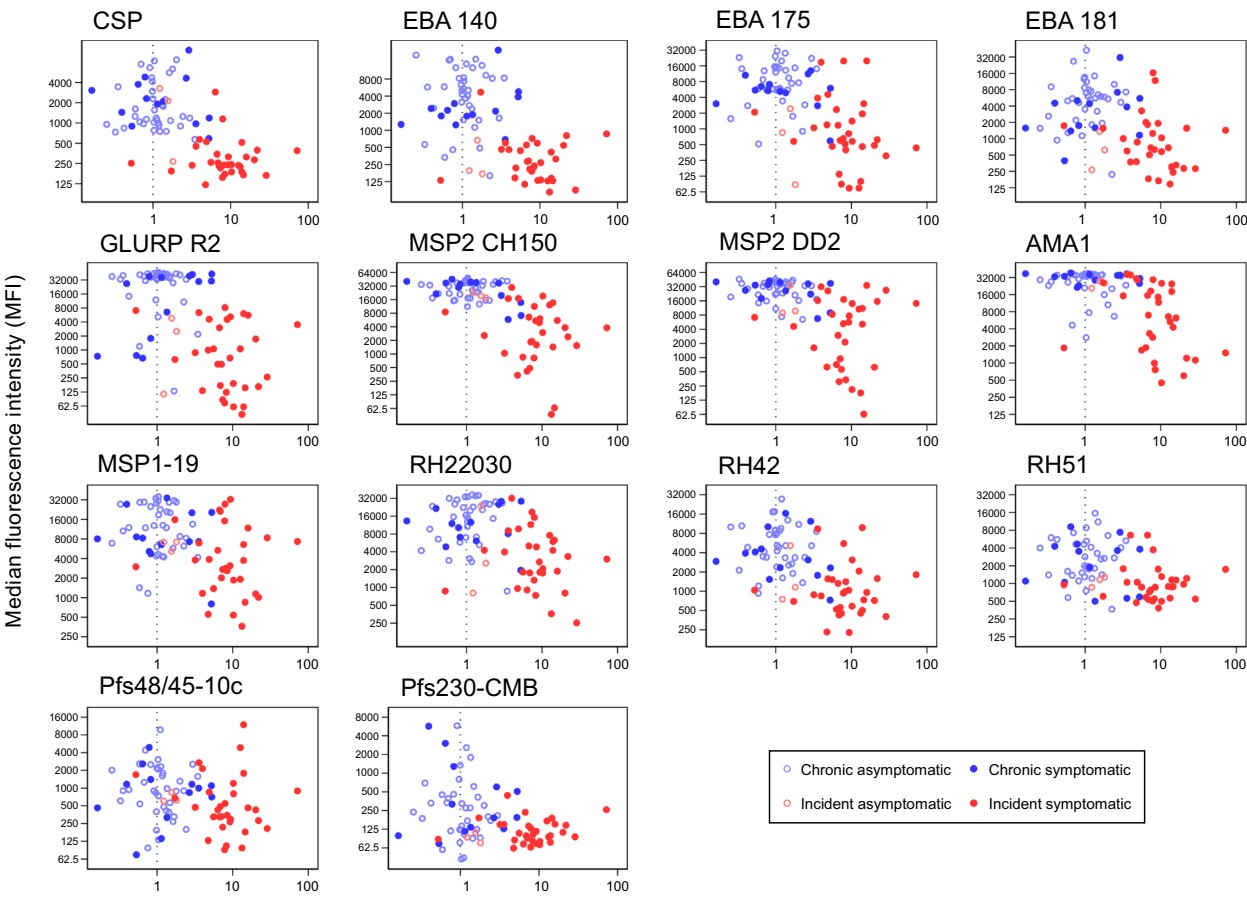

**Fig. 6 Magnitude of asexual and sexual stage antibody response and parasite multiplication rate.** Magnitude of antibody response to specific *P. falciparum* antigens is given as the background-adjusted median fluorescence intensity (MFI). Parasite multiplication rate (PMR) is presented as the geometric mean of each individuals PMR observations, with a dashed line at PMR = 1 (no change in parasite density over 48 h intervals). Plasma samples were taken from the start of intensive follow-up for both cohorts (incident *n* = 48, chronic *n* = 51). Light blue hollow markers = chronic asymptomatic infection, dark blue solid markers = chronic symptomatic infection, light red hollow markers = incident asymptomatic infection, dark red solid markers = incident symptomatic infection. Full details of all antigens are in Supplementary Table 1.

malaria symptoms that may require treatment. Finger-prick samples were taken on a daily basis for 7 days. After this first week, sampling was performed weekly until symptoms occurred (measured temperature ≥37.5 °C or reported fever in last 24 h) or, if still without symptoms, day 35. Artemether-lumefantrine treatment (AL; Coartem; Novartis Pharma) was given when symptoms occurred or on day 35, whichever came first. Upon the first detection of infection, participants were invited to the insectary for assessment of infectivity; a second feed was performed on the first day of symptoms or on day 14 following the first detection of parasites (whichever came first). For individuals who remained asymptomatic throughout follow-up, a third and final membrane feeding assay was performed on day 35. The use of nPCR during screening and for infection confirmation was important for rapid sample turn-around and was performed on-site. Higher sensitivity qPCR assays (also 18s-based[58]) were performed later at Radboudumc, and the results of these assays supersede those of nPCR for all analyses.

**Sampling: Chronic infection cohort.** During screening for the chronic infection cohort, individuals were eligible for enrolment if all other eligibility criteria were met regardless of parasite status. Monthly sampling by nPCR was performed 3–4 times, up to the peak of the transmission season. RDT's were performed concurrently with all sampling for nPCR; a positive RDT accompanied by temperature of ≥37.5 °C prompted immediate treatment with AL as for incident infections. Individuals were defined as having chronic asymptomatic infection when parasites were detected by nPCR at two consecutive sampling visits, spaced approximately 1 month apart, without reported illness, accompanying fever or other apparent malaria symptoms. These individuals were invited to participate in more intensive sampling. This intensive sampling phase was identical to the follow-up described after infection detection in the incident infection cohort, starting 1–2 days after the monthly visit that confirmed eligibility.

**Mosquito feeding assays.** Membrane feeding was performed as described in detail in our online protocol to determine infectivity to locally reared female Anopheles *gambiae ss* mosquitoes[59]. Briefly, heparinized blood collected in vacutainers was immediately transferred to a glass mini-feeder and offered to 50–80 mosquitoes. Mosquitoes were kept at 27–29 °C in the insectary on glucose and dissected 7 days after feeding. Mosquito midguts were examined for the presence of parasite developmental stages, oocysts, by two independent microscopists.

**Parasite detection, quantification and complexity of infection.** Thick blood films were stained with Giemsa and independently read by expert research microscopists over 500 fields for quantification of gametocytes and asexual parasites. The immediate molecular detection of parasites by qualitative *18S*-based nested PCR (nPCR) was done at CNRFP[26], Ouagadougou for the detection of incident and chronic infections. DNA was extracted from 20 µL of blood collected on filter paper. DNA extraction was undertaken using Qiagen QIAamp blood extraction kits as per the manufacturer's instructions (Cat No./ID: 51306). Confirmatory *18S*-based quantitative real-time polymerase chain reaction (qPCR) was performed at Radboudumc the Netherlands; parasite DNA was extracted from whole blood, with a detection limit of ~0.01 parasites/µL of blood (equating to 1 parasite in each 100 µL sample of extracted blood)[58]. For subsequent analysis, total nucleic acids were extracted from EDTA blood stored in RNA protect cell reagent (Qiagen, Hilden, Germany) from 100 µL blood samples stored at CNRFP at −80 °C until shipment on dry ice. Extraction was done using a MagNAPure LC automated extractor (Total Nucleic Acid Isolation Kit-High Performance; Roche Applied Science, Indianapolis, IN, USA). Gametocyte quantification with qRT-PCR amplifying female (*Pfs25*; PF3D7_1031000) and male (*PfMGET*; Pf3D7_1469900) gametocyte mRNA was done using sex-specific trendlines of cultured gametocytes of the PfDynGFP/P47mCherry reporter line[60]. Samples were declared gametocyte negative if the estimated

gametocyte density was <0.01/μL (1 gametocyte per 100 μL blood sample); estimates of male and female gametocytes were adjusted for background signal from asexual parasites[61]. qPCR targeting *pfap2-g*[62], *gexp-5*[63] and *sbp1*[64] was performed and the ratio of *ap2-g* or *gexp-5* copies to ring-stage parasite density (extrapolated from *sbp1* copies) was used as a measure of the proportion of sexually committed ring-stage parasites[45]. Primer sequences are provided in Supplementary Table 4.

The complexity of infection (COI) was determined by targeted deep sequencing of apical membrane antigen 1 (AMA1)[65]. COI was calculated for samples collected on day 0, 1, 14 and 35. COI from samples from day 0 or 1 were averaged to provide a single baseline COI. Human haemoglobin S (HBs) and C (HBc) were genotyped by DNA amplification by multiplex PCR, followed by allele-specific primer extension with SNP-specific probes and hybridization to magnetic beads[66].

**Immuno-assays.** IgG antibodies against 14 antigens, 1 targeting pre-erythrocytic stages (Circumsporozoite protein [CSP][67]), 11 targeting the asexual blood stage (Erythrocyte binding antigen [EBA140, EBA175 and EBA-181][34]; Glutamate rich protein 2 [GLURP-R2][68]; Merozoite surface protein 1–19 [MSP1-19][69], Merozoite surface protein 2 [MSP2-ch150/9 (3D7 family allele)][37], and MSP2-DD2 (FC27 family allele)][36]; Apical membrane antigen 1 [AMA1][70], and Reticulocyte binding protein homologue [RH2.2[71], RH4.2[72], RH5.1[73]]), and 2 targeting sexual stages (Pfs48/45-10c[74]; Pfs230-CMB[75]) were quantified at baseline for each participant using a Luminex MAGPIX© suspension bead array[76]. Serological analysis was conducted on serum samples collected at the start of intensive follow-up (day 0) (incident cohort $n = 48$, chronic cohort $n = 51$). Serum was assayed at a dilution of 1:200. Secondary antibody was an R-phycoerythrin conjugated goat anti-human IgG (Jackson Immuno Research, PA, USA; 109-116-098) diluted to 1:200. Data are presented as median fluorescence intensity (MFI).

**Data analysis.** All data were recorded on paper forms, double entered in Microsoft Access 2010 and imported into Stata 15.0 (StataCorp LP, Texas, USA). Peak parasite densities were compared between the cohorts using Wilcoxon's rank-sum test. Changes in parasite densities over time were assessed by mixed-effects linear regression on log parasite densities, with random effects for participants and a linear effect for time. Total parasite multiplication rate (PMR) was compared between different groups using mixed-effects linear regression on log PMR, with random effects for participants; PMR itself is based on biological assay readouts (18s qPCR based parasite densities) and was observed to exceed 32 (the theoretical maximum PMR in one erythrocytic cycle) for a number of samples. Correlations were calculated using Spearman's rank correlation (presented as Spearman rho). Associations with day 14 gametocyte densities were calculated using Tobit regression on log-transformed densities, with a cut off at 0.01 gametocytes per microliter. Differences in infectivity to mosquitoes were assessed by binary regression with a log link. Gametocyte density was accounted for using a model previously described[24]. Repeated observations were taken into account using robust standard errors. Cohort (i.e. incident or chronic infection) was included in the regression analysis of infectivity along with symptoms (where symptoms were concurrent or present at any point during follow-up). The relationship between time to symptoms and grouped antibody response was assessed with cox regression separately for each cohort. Study participants were divided into tertiles of response to all pre-erythrocytic and asexual stage antigens based on the first component of a principal components analysis (PCA) of the log-transformed MFI. For the analysis of sexual stage antibody response and infectivity, tertiles of response to pre-erythrocytic and asexual antibodies were included as covariates in the regression model, along with total gametocyte density as determined at the same time point.

**Reporting summary.** Further information on research design is available in the Nature Research Reporting Summary linked to this article.

## Data availability

All data associated with this study are provided in the online Dryad repository (https://doi.org/10.5061/dryad.pc866t1n3). Source data are provided with this paper.

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

## Acknowledgements

This work was supported by a fellowship from the European Research Council (ERC-2014-StG 639776), the Bill and Melinda Gates Foundation (INDIE OPP1173572) and the Radboud-Glasgow Collaboration Fund. J.B. received support from the UK MRC and the UK DFID (#MR/R010161/1) under the MRC/DFID Concordat agreement and as part of the EDCTP2 Programme supported by the European Union. W.S is funded on a Sir Henry Wellcome fellowship (number 218676/Z/19/Z) from the Wellcome Trust, UK. Recombinant proteins were kindly provided by Simon Draper (Rh5.1), James Beeson (EBA140, 175, 181, Rh2_2030 and Rh4.2), Susheel Singh (GLURP-R2),

Tony Holder (MSP1-19), Mike Blackman (AMA1), Eleanor Riley (MSP2-DD2), Kevin Tetteh (MSP2 CH150/9), Michael Thiesen/Susheel Singh (Pfs4845-10c), or Fraunhofer (Pfs230-CMB).

## Author contributions
Study design: C.D., A.B.T. and T.B.; data collection: A.B., W.S., A.O., I.S., N.I.O., S.S. S., L.G., C.P, L.W., J.J.B., O.J., S.S.A. M.O., C.W.T., D.K., Z.S. and S.B.S.; data analyses and interpretation: A.B.T., J.B., W.S., M.M., C.D, A.B.T. and T.B.; statistical analysis: J.B. W.S.; paper writing and critical review: A.B., J.B., W.S., M.M. C.D., A.B.T. and T.B. All authors have contributed to and approved the final version of the manuscript.

## Competing interests
The authors declare no competing interests.
