## [Peer Review File · Nature Communications]

Reviewer comments, first round –

Reviewer #1 (Remarks to the Author):

This MS presents a study describing the longitudinal follow-up of two cohorts in Burkina Faso and investigates transmission stage production and infectivity to mosquito in chronic and acute malaria infections. The data appear of good quality and there are some useful observations. Specifically the observation that ap2g is linked to later development of mature gametocytes is a welcome confirmation of ap2g as a biomarker of commitment. Whilst recognising that the data presented is significant and represents a big undertaking, there are several aspects which dampen the quality and the ambition of the manuscript, these are detailed below:

1-The evaluation of genetic diversity of the infections by a single polymorphic marker. On top of being a fairly low resolution method of evaluating genetic diversity of the parasite population, this appears to be only evaluated at day 1, 2 and 3 of follow-up. How do the authors account for the possibility of new infections, this is especially relevant in the C group (chronic, remained asymptomatic), and how this may impact the conclusions of the paper.

2-The observations of antibody levels and their relation to PMR are of interest but appear to be quite separate from the main message of the paper and may be better suited to be presented as a separate paper.

3-The authors state in the discussion : "This may be due to chronic inflammation and hence reduced levels of Lysophosphatidylcholine". Why did the authors not measure lysoPC levels directly in plasma? This seems like a significant missed opportunity and this observation would really bolster the impact of the paper.

4-The correlation between AP2-g level and D14 gam/D0 ring whilst convincing is moderate. Whilst not doubting this expected association, it would be very useful to test and validate other markers (e.g. gdv1, gexp05) to see if they perform better.

5-The quality and readability of the figures is quite poor (see below) and the quality of the writing is very uneven .

Minor points and corrections.

The abstract is overly descriptive

Fig1- the inclusion of d5 and d17 as examples is a bit confusing. Some items are too small to be readable.

Fig 2- lines and endpoints are not discernable in A and B, these panels are not very informative - legend 'x' instead of 'y'

Fig4 - would be better to also display and test incident symptomatic alone in C

Fig s2 Numbers of symptomatics don't match those in the text

There are quite a number of inaccuracies or inconsistencies throughout the text, eg. Some supplemental items are not mentioned in the main text (f1, t2)

p13 figure 4A -> 5A

p7 PMR instead of MR

Reviewer #2 (Remarks to the Author):

This is a well-designed study that addresses an important knowledge-gap in our understanding of malaria transmission. There are some concerns about the execution, reporting and analysis presented here that weaken the main conclusions.

Major issues

1. Immunology (serology) is disjointed/disconnected from rest of story.

a. It doesn't add to what we learn particularly since it is only treated cross-sectionally. It is not

mentioned in the abstract which further suggests it is not important.

b. In addition, there were no sexual-stage antigens investigated which limits the ability to relate this information to the main experiment.

c. Are any of the antibody results correlated with time until symptoms?

2. Feeding results – there were very few infected mosquitos particularly in the incident cohort. The data is not presented in a way that allows one to evaluate the contribution of individuals to these results.

a. Are feeding successes within individuals correlated?

b. Why no feeding post tx in the incident cohort? Studies by some of these authors indicated significant potential to infect mosquitoes post Tx.

c. Is feeding success in the incident cohort associated with time to symptoms? Maybe too few feeding successes to analyze this.

d. Given that feeding success was relatively rare and that “there was considerable variation between individuals in gametocyte densities, densities within individuals were relatively stable”, the authors should look more at individual heterogeneity.

e. The feeding results are presented without complete details. How many participants are represented in the 212 feeding assays? How many participants resulted in infected mosquitoes? Did the 5 infected mosquitoes come from one person in the incident cohort? Given that there were only 5 infected incident mosquitoes, this seems a very small sample size from which to draw conclusions.

f. What does this statement mean? “After adjusting for gametocyte density and cohort, the risk of infecting a mosquito at any point was not lower in infections that became symptomatic.” This seems to contradict the stated take-away message that symptomatic infections were less infectious.

g. The use of ‘no significant correlation’ is somewhat misleading or too strong since the sample size was (presumably) very small and the problem might well be power to detect such an effect.

h. Supplemental Fig 1 – many participants did not have feeds and this is not well explained or clear in the main text. Also, why were there no feeds on the days that chronic infections became symptomatic?

i. It would be a stronger comparison to look at the infectiousness of chronic individuals on days without symptoms and days with symptoms

3. The authors are trying to make an argument that AP2G copies at day 0 are correlated to gametocytes at d14/rings d0. This seems to be true for chronic infections but not for incident infections. Furthermore, d0 for a chronic infection is not any particular day in the course of their current infection, it just marks the beginning of monitoring so all that can be said from this data is that AP2G copies seem to track with gametocyte density 14 days later in chronic infections. However, this correlation is not rooted in any other context or comparison. Maybe AP2G copies correlate with gametocyte ratio on any day or even on prior days.

Minor

1. Abstract. Need to mention in the abstract that the observations here may be related to gametocyte maturation.

2. Abstract. The last sentence in the abstract is somewhat vague and lacking impact. Yes, we should continue to improve timely case management and screen for infections but we don't need this study to tell us so.

3. Methods. The ethics statement is confusing. Why are they screening individuals over 12 years if the age range is 5-10 years. Who is providing consent for screening? A 12-year old? Make this clearer

4. Methods. Be very clear about which measurements were taken with which kind of sample. I can see that finger-stick samples were taken (based on statement later in the methods about extraction from filter paper) but this should be explicit earlier (Sampling section). Later, it mentions qPCR done with whole blood – so was venous blood taken at every visit? In addition, since mosquito feeding surely required venous blood draw, it should be clear which molecular assays are done with venous blood and which are done with finger sticks. Did you use finger-stick and venous blood measurements when calculating PMR?

5. Results. Use enrollment and recruitment consistently. I'm confused in Table 1 where

recruitment density is same between chronic and incident cohort. I think this means density at day 0 for chronic but density at day of treatment for incident? Or does it mean density at day 0 for both? Is day 0 recruitment or enrolment or neither?

6. Results. Indicate how density was measured in Table 1. I assume by qPCR

7. Results. Following the comment above, were these individuals similar at screening – ie parasite density by qPCR? In other words, were these cohorts essentially the same at screening and then only differentiated by presumptive tx? How many of the chronic cohort had microscopically-detectable parasites?

8. Results. Following the comment above, it would have been interesting to see what would have happened if the chronic cohort, after completing the followup, was treated and then followed for incident infection. Are these groups essentially identical and the differences in PMR, gametocytes and infectivity related to acquisition of a new infection or perhaps just the age of the infection/maturation of gametocytes?

9. Results. The red line in Figure 2B that shoots up doesn't have a black dot but stops at about 10-12 days. Who is this? Not one of the incident cohort that had no symptoms by Day 35. Is the dot missing or was this person lost to followup? I guess this is the same participant as the lower left panel in supplemental figure 2A?

10. Discussion. The statement that asymptomatic individuals contribute more to the reservoir is too strong given the very limited age range in this study.

11. Supplemental Fig 1 – shouldn't everyone in B have a red dot?

12. Supplemental Fig 2. Why do some participant's trajectories stop before day 35 with no red dot (supplemental fig 2B)

Very minor/grammar or style edits

1. First two sentences of the intro start with describing something as "poorly understood"
2. Methods, Data Analysis: changed should be changes

Reviewer #3 (Remarks to the Author):

This is a fascinating report of an intensive surveillance study of the dynamics of infection and infectiousness in a cohort of school-aged children in Burkina Faso.

The scientific methods are sound a rigorous. However, I struggle to understand the specific questions and/or hypotheses being addressed and wonder if the findings of this study are generalizable.

My specific major concerns are:

1. Incident and chronic infections were treated differently. Children with incident infections were treated with long-acting dihydroartemisinin-piperaquine which may have impacted the subsequent infection dynamics and infectiousness (especially as the median time to infection was relatively brief).
2. It is difficult to disentangle the host vs. parasites vs. host-parasite interaction factors that contributed to children becoming incident vs. chronic infections. Thus, the studies many parameters (antibody level, PMR, ap2-g level) highlight associations but offer no insight into causation.
3. I may have misunderstood, but the timing of the mosquito infectivity assay was not standardized across cohorts.
4. The figures provide details on every measurement but it remains unclear what the conclusions are with so many data points.
5. The manuscript does not explain why these specific outcomes were selected, as the introduction seems to focus on symptomatic status and gametocyte dynamics.

I have identified several specific areas that could be addressed:

Abstract

- Would be helpful to define incident vs. chronic cohort
- Define prospective gametocyte production

Results

Incident infections occurred with a median of ~4 weeks. What occurred during the follow up time

after the incident infection?

Figure 1 is very detailed and difficult to follow. A more simplified depiction of the study design might help the reader to understand.

There was no mention of the relationship between gametocyte sex ratio and infectiousness. This could be an interesting piece of data.

Table 2 presents coefficients that are difficult to interpret. What was the baseline distribution of hemoglobin type?

Mosquito infectivity assays: A series of bivariate analyses are presented (chronic vs incident, symptomatic vs asymptomatic), but it is not clear if these contribute independently or there is confounding.

Methods

Please clarify the reason for differentiating qPCR and nPCR?

Mosquito feeding assays were performed without serum replacement. Were sexual stage serological responses assessed? Only erythrocytic antigens were presented though a major focus on the manuscript was gametocyte production and infectiousness.

REVIEWER COMMENTS

Reviewer #1 (Remarks to the Author):

This MS presents a study describing the longitudinal follow-up of two cohorts in Burkina Faso and investigates transmission stage production and infectivity to mosquito in chronic and acute malaria infections. The data appear of good quality and there are some useful observations. Specifically, the observation that ap2g is linked to later development of mature gametocytes is a welcome confirmation of ap2g as a biomarker of commitment. Whilst recognising that the data presented is significant and represents a big undertaking, there are several aspects which dampen the quality and the ambition of the manuscript, these are detailed below:

1-The evaluation of genetic diversity of the infections by a single polymorphic marker. On top of being a fairly low resolution method of evaluating genetic diversity of the parasite population, this appears to be only evaluated at day 1, 2 and 3 of follow-up. How do the authors account for the possibility of new infections, this is especially relevant in the C group (chronic, remained asymptomatic), and how this may impact the conclusions of the paper.

Whilst MSP2 as polymorphic marker has been used successfully to measure the force of infection (e.g. Mueller et al. doi: 10.1073/pnas.1200841109), it may indeed not capture all transmissible parasite clones (Grignard et al. 10.1016/j.ijpara.2018.02.005) and is prone to preferentially amplifying smaller products. As part of the revision, we have therefore performed amplicon deep sequencing of the polymorphic antigen Apical Membrane Antigen 1 (AMA-1). Amplicon deep-sequencing provides better sensitivity to detect minority clones than microsatellites or length-dependent polymorphisms such as MSP-2. Moreover, we have expanded the number of days at which we determined the complexity of infection. We now also included days 14 and 35 when mosquito feeding was performed to, as the reviewer suggested, ensure we take into consideration newly acquired infections. Whilst MSP-2 and AMA-1 data showed very strong correlation in terms of estimated complexity of infection (COI) ($\rho = 0.58$, $p < 0.001$) and we confirmed that COI at enrolment was not associated with parasite multiplication rates ($p = 0.139$), the suggestion to assess COI at later time-points was very valuable and improved our analyses.

Using the revised approach, we show that the complexity of infection among infections that remained asymptomatic for the duration of follow-up stayed almost identical. COI was greater in chronic infections, and was positively associated with infection outcome (higher COI was associated with higher proportion of infected mosquitoes), after controlling for gametocyte density. We have adjusted this section of the results as follows:

'Deep sequencing of apical membrane antigen 1 (AMA1) demonstrated that the majority of cohort participants had clonally complex infections at the start of intensive follow-up, both for incident infections (67% [31/46] multi-clonal, median COI 3 [IQR 1-4]) and chronic infections (93% [51/55] multi-clonal, median COI 6 [IQR 5-9]) (table 1). The prevalence ($p = 0.001$) and complexity ($p < 0.0001$) of multi-clonality were higher in chronic infections than incident infections at the start of follow-up. Among the few incident infections that remained asymptomatic, 33% (2/6) and 100% (2/2) were multi-clonal at days 14 and 35 respectively. Chronic infections showed similar complexity throughout follow-up (day 14 = 98% multi-clonal [46/47], median COI 6 [IQR 4-8], day 35 = 95% multi-clonal [39/41], median COI 6 [IQR 4-8]). Neither multiplicity of infection at the start of follow-up ($p = 0.139$) nor Hb genotype ($p = 0.914$) were associated with PMR. After controlling for gametocyte density and cohort, compared to those with 3 or fewer clones, those with 4-6 clones were 4.23 [95% CI 1.37-13.10] times more likely to infect a mosquito, and those with 7 or more clones were 4.30 [95% CI 1.73-10.71] more likely ($p = 0.007$).'

2-The observations of antibody levels and their relation to PMR are of interest but appear to be quite separate from the main message of the paper and may be better suited to be presented as a separate paper.

In the revised manuscript, we included gametocyte antigens Pfs48/45 and Pfs230 in our panel of Plasmodium antigens for serology. This improves the coherence of the manuscript and brings novel insights in the importance of naturally acquired immune responses on gametocyte infectivity.

Since antibody responses are acquired with age and modulate PMR, and there are several previous indications for a role for age-dependent acquisition of anti-gametocyte immune responses in gametocyte production and infectivity (Stone et al. doi: 10.1038/s41467-017-02646-2; Dantzer et al. doi: 10.1126/scitranslmed.aav3963), we consider it of importance to retain our data on antibody levels and PMR. With the inclusion of Pfs48/45 and Pfs230, we believe these data are no longer separate from the main message of the paper.

3-The authors state in the discussion: “This may be due to chronic inflammation and hence reduced levels of LysoPC levels directly in plasma? This seems like a significant missed opportunity and this observation would really bolster the impact of the paper.

This is an interesting suggestion based on earlier work of the group of one of the co-authors (Brancucci et al. doi: 10.1016/j.cell.2017.10.020). It is, however, far from straightforward. We are not yet in the position to draw any conclusions from such data even if we had them. There is no *in vivo* data correlating LysoPC levels, inflammation and gametocyte production. Given the complexity of such interactions, these would best be assessed in controlled human infections. Importantly, we do not know yet the contribution of systemic versus hematopoietic infection to gametocyte production. This is an important knowledge gap that should be addressed before exploring the implications of systematic inflammation to gametocyte commitment. We are currently addressing these issues that are major undertakings and beyond the scope of this manuscript.

We have included a statement on these future studies in the revised discussion section.

‘There is currently no in vivo data correlating LysoPC levels to inflammation and gametocyte production. Following future assessments of the contribution of systemic versus hematopoietic infection to gametocyte production, the implications of systematic inflammation to gametocyte commitment can be examined.’

4-The correlation between AP2-g level and D14 gam/D0 ring whilst convincing is moderate. Whilst not doubting this expected association, it would be very useful to test and validate other markers (e.g. gdv1, gexp05) to see if they perform better.

We acknowledge that our observations, although novel and in our view exciting, are not presented as strongly as they could be. We appreciate the suggestions that allowed us to add valuable new data to our revised manuscript. As a first step to strengthening our case, we performed an analysis requested by reviewer 2 using the ap2-g/ring stage parasite ring ratio in a period between day 0 and day 14, demonstrating that the correlation we observed is not present between non biologically coherent time points:

‘The ratio of gametocytes/ μ L on day 14 to rings/ μ L on day 0 (i.e. the proportion of baseline asexual parasites that subsequently produced gametocytes) was significantly associated with the ratio of ap2-g transcript copy number to ring stage parasite density on day 0 (i.e. the relative abundance of ap2-g) (Spearman rho 0.42, $p=0.0015$) (figure 5A). This association was not observed when gametocyte production at day 14 was examined in relation to ap2-g production 6-8 days prior (Spearman rho = 0.04, $p=0.780$), an interval that is too short to allow gametocyte production.’

As suggested by reviewer 1, we also performed new qRT-PCR assays to test another marker – GEXP-5, for its association with gametocyte productivity. Performing these additional molecular assays resulted in a delay in submission, caused by the shortage of extraction kits and other molecular reagents in the present time, but significantly improved our revised manuscript. The marker suggested by reviewer 1 was even more strongly associated with gametocyte production and confirmed our findings with ap2g on lower gametocyte commitment in incident infections.

'An independent marker of early sexual stage development, Gametocyte exported protein 5 mRNA (gexp-5), which is detectable in the cell cytoplasm within 14h of gametocyte formation [28], was strongly associated with gametocyte production (Spearman correlation coefficient 0.70, $p < 0.0001$), and was significantly higher in chronic infections ($p < 0.0001$). Relative ap2-g and gexp-5 transcript abundance were correlated (Spearman rho 0.55, $p < 0.0001$). The associations of ap2-g and gexp-5 transcript abundance with gametocyte conversion rate remained highly significant after adjustment for gametocyte density at the time of measurement ($p < 0.0001$).

In addressing the reviewers concerns over our presentation of ap2-g data, we have decided to remove supplemental figure 4. This figure showed the general spread of ap2g/ring ratio over time-points in the first week of intensive follow up, but was not discussed in the results section in the first draft. We have concentrated on d0 values because of their biological relevance to gametocyte density measured 14 days later. To simplify our messaging, we thought it appropriate to remove the figure *in toto*.

5-The quality and readability of the figures is quite poor (see below) and the quality of the writing is very uneven.

We have edited our draft thoroughly, and addressed all specific points on style/errors picked up by the reviewers. We have also redrawn many figures, and re-written the results to improve structure, more clearly articulate our hypotheses and better bring across the study findings and their implications.

In the introduction, we now state:

'We hypothesized that a large proportion of incident infections would develop into chronic asymptomatic infections that are highly infectious to mosquitoes and that both host and parasite factors influence gametocyte production and infectivity. To test these hypotheses, we recruited...'

Minor points and corrections.

The abstract is overly descriptive

Fig1- the inclusion of d5 and d17 as examples is a bit confusing. Some items are too small to be readable.

Figure 1 has been simplified as proposed.

Fig 2- lines and endpoints are not discernible in A and B, these panels are not very informative - legend 'x' instead of 'y'

We appreciate the reviewers' feedback on figures 2A and B, and accept that the data presented is rather dense. We have therefore decided to move 2B to the supplemental data. The intent for 2A was not to allow the reader to follow the trajectory of every individual's parasite density – supplemental figure1 does this. Instead we wanted a graph to clearly demonstrate the variability of these trajectories within each cohort, and the stark differences between the cohorts. We believe the revised figure 2A achieves this goal.

Fig4 - would be better to also display and test incident symptomatic alone in C

We agree that our findings would be stronger if we were able to plot and test the association in figure 4C for each cohort. However, only 4 observations from this cohort reached day 14 during follow-up without developing symptoms and had ap2g measured. Therefore, only for a very small number of individuals in the incident infection cohort were we able to estimate the d0/d14 gametocyte conversion. We chose to ignore cohort in this analysis for this reason, and decided to keep these data points in the graph for transparency. Our main findings, that ap2g and gexp5 transcripts were associated with gametocyte production and that ap2g and gexp5 transcripts are less abundant in incident infections, are evident from the revised figure 5.

Fig s2 Numbers of symptomatics don't match those in the text

This is true – we had overlooked this. 2 symptomatic markers (red dots) for incident infections and 1 for chronic infection are missing from the plot. The reason for this is that these individuals became symptomatic at time points without a molecular parasite/gametocyte density measurement. We have now addressed this in the figure legend explicitly, giving details of the 3 missing spots, and have added asterisks in the appropriate plots to make clear when these individuals became symptomatic.

There are quite a number of inaccuracies or inconsistencies throughout the text, eg. Some supplemental items are not mentioned in the main text (f1, t2)

p13 figure 4A -> 5A

p7 PMR instead of MR

We thank the reviewer for pointing out these errors. Figure S1 and supplemental figure 1 and table 2 are now mentioned in the text as follows:

'Screening and follow-up time-points are shown for all individuals in supplemental figure 1.'

'Details of the association of specific antigens with PMR and symptomatic status are shown in supplemental table 2.'

Parasite PMR is now referred to with consistent language throughout the report. The error on page 13 has been remedied.

Reviewer #2 (Remarks to the Author):

This is a well-designed study that addresses an important knowledge-gap in our understanding of malaria transmission. There are some concerns about the execution, reporting and analysis presented here that weaken the main conclusions.

Major issues

1. Immunology (serology) is disjointed/disconnected from rest of story.

a. It doesn't add to what we learn particularly since it is only treated cross-sectionally. It is not mentioned in the abstract which further suggests it is not important.

b. In addition, there were no sexual-stage antigens investigated which limits the ability to relate this information to the main experiment.

We thank the reviewer for their comments. Suggestion b from reviewer 2 has allowed us to add an important finding to our manuscript. By testing responses to gametocyte/gamete antigens Pfs230

and Pfs48/45, we i) solved the apparent disconnect between the immunology and the remainder of the paper; ii) demonstrated an important effect of responses against these antigens on gametocyte infectivity. Infectivity was 65% (95% CI 21-85) lower for a 10 fold increase in anti-Pfs48/45 antibody concentration ($p=0.011$); for Pfs230, infectivity was 64% (95% CI 34-80) lower for a 10 fold increase. These findings now receive prominence in the revised abstract and results section.

Our decision to keep our immunological analysis cross sectional was principally due to the short duration of follow up, and disparity in duration of follow-up between cohorts. Our aim was to provide an indication of the immune environment present at the start incident infections, or at the point at which chronic infections were confirmed. The generally shorter follow-up of most incident infections would have resulted in a severe limitation when comparing cohorts. Longitudinal analysis would have been possible for most chronic infections, but the added value of additional time-points would be minimal, as longitudinal data for most antigens have been reported elsewhere. Though we believe our cross sectional approach is the right one in the circumstances, the reviewers comments led us to reassess the time-point chosen for serological analysis of incident infections; previously we used samples collected at the start of weekly screening for incident infections (an average of 4 weeks before incident infection was confirmed). We now use blood samples taken at the point of incident infection confirmation, which we feel better reflects the immune context of the infection.

c. Are any of the antibody results correlated with time until symptoms?

The reviewer raises an interesting point that we had not considered in our original submission. Responses to two antigens were indeed associated with a reduced likelihood of experiencing symptoms in the incident cohort. These data are incorporated in the revised manuscript and supplemental table 2.

2. Feeding results – there were very few infected mosquitos particularly in the incident cohort. The data is not presented in a way that allows one to evaluate the contribution of individuals to these results.

The reviewer is correct that a low proportion of mosquitos became infected in the incident cohort. This is in fact one of the main findings of our study. We agree that the summarized results in figures 5 A and B do not provide detail of the individual infections. Also the 1st reviewer requested additional information. We have therefore added a table to the supplemental data (supplemental table 3) that gives more details of the mosquito infections in each cohort.

a. Are feeding successes within individuals correlated?

This is a valuable comment that we had not addressed in our first draft. During our revision, we have analysed our data to examine within individual correlations of infectivity outcomes. We have added the following line to the results section to address individual level heterogeneity in mosquito infection rate.

'As demonstrated previously, there was a positive association between gametocyte density and mosquito infection rate (figure 5A, Spearman correlation 0.51, $p<0.0001$)²⁴. Given that gametocyte densities are variable between and stable within individuals it is unsurprising that there is also heterogeneity in infectivity between individuals ($p<0.0001$). After adjusting for gametocyte density, there was still strong evidence for heterogeneity in infectivity between individuals ($p<0.0001$).'

In response to the reviewer's comments above we have also gone a step further toward identifying factors that may contribute to heterogeneity in infectivity at the individual level, with an analysis of

specifically sexual stage (transmission blocking) antibody responses. This plausible reason for heterogeneity between individuals is included in the revised discussion section.

b. Why no feeding post tx in the incident cohort? Studies by some of these authors indicated significant potential to infect mosquitoes post Tx.

The reviewer is correct that we have shown that mosquito infection can persist after ACT treatment. The effect of most ACTs (including those in this study) on asexual parasites, gametocytes and infectivity is well established, and though it would have been interesting to observe variations in their effect with infection presentation, this was not judged to add significant value to our study. To limit the number of venous bleeds in this young study population and avoid the complexity of phlebotomy near the insectary for clinical cases that require immediate treatment, we therefore decided in our protocol to perform mosquito feeding only pre-Tx.

An examination of post-treatment infectivity with sufficient power for meaningful interpretations would have required a different study design. During the revision of our manuscript, some of the co-authors have published an individual-level meta-analysis of transmission following treatment with ACTs with and without primaquine (Stepniewska et al., doi: 10.1093/infdis/jiaa498) that demonstrated that relatively few infected mosquitoes can be expected after artemether-lumefantrine whilst transmission is more common after dihydroartemisinin-piperazine. Dedicated studies like the WWARN meta-analysis are more powerful to quantify post-Tx transmission potential than observational studies like the current study.

c. Is feeding success in the incident cohort associated with time to symptoms? Maybe too few feeding successes to analyze this.

The reviewer is correct – there were too few incident infections with successful mosquito infection; 5 infected mosquitoes from 2 individuals. The 2 individuals who infected mosquitoes were infectious on days 1 and 16 after start of the intensive follow-up period. This information is now added to the revised results section. The association between time since enrolment and infectivity to mosquitoes was tested in chronic infections. We have added a line to the results to highlight why incident infection feeds were not studied in more detail.

‘During chronic infections there was no significant effect of assay timing (days after enrolment) on infectivity, wether adjusted ($p=0.332$) or unadjusted ($p=0.058$) for gametocyte density (figure 5B); too few incident infections gave rise to mosquito infection to allow the same analysis but the two infectious feeds in this cohort were on days 1 and 16 after enrolment.’

d. Given that feeding success was relatively rare and that “there was considerable variation between individuals in gametocyte densities, densities within individuals were relatively stable”, the authors should look more at individual heterogeneity.

We thank the reviewer for highlighting this. We believe the point has been addressed in response to point a above. We would like to stress that feeding success was in line or higher than in previous studies. The fact that 26/60 chronic infections (46.7%) resulted in at least one infected mosquito suggests that our repeated infectivity assessments leads to higher estimates of transmission potential compared to previous assessments where 5-10% of asymptomatic infections was infectious (Goncalves et al. doi: 10.1038/s41467-017-01270-4; Tadesse et al. doi: 10.1093/cid/cix1123.).

e. The feeding results are presented without complete details. How many participants are represented in the 212 feeding assays? How many participants resulted in infected mosquitoes?

Did the 5 infected mosquitoes come from one person in the incident cohort? Given that there were only 5 infected incident mosquitoes, this seems a very small sample size from which to draw conclusions.

We agree that our original submission did not have a lot of detail on the individual feeding experiments. We have thus added a supplemental table with details of mosquito infection experiments and the following text in the Results section:

'2/34 individuals with incident infection were infectious to mosquitoes, resulting in 0.1% (5/3,403) of mosquitoes becoming infected overall, compared to 28/60 individuals with chronic infection who infected 4.5% (554/12,405) of mosquitoes.'

We disagree that these are small numbers. The denominator is large for examination of the proportion of infected mosquitoes. Although few incident infections resulted in infected mosquitoes, this is based on a large number of examined mosquitoes and thus a fairly precise estimate. Clearly, very few comparisons can be made within the group of incident infections but the comparisons with chronic infections have sufficient power. We therefore feel that our conclusions strike the right balance; surety in our analysis of inter-cohort differences (e.g. lower infection rates were likely due to rapid symptom development and low gametocyte density), caution in our analysis of infection rates within the incident cohort (e.g. Infection rate's association with gametocyte density, time to symptoms etc).

f. What does this statement mean? "After adjusting for gametocyte density and cohort, the risk of infecting a mosquito at any point was not lower in infections that became symptomatic." This seems to contradict the stated take-away message that symptomatic infections were less infectious.

This analysis considers the effect of symptoms on infectivity regardless of cohort, but cohort was nonetheless included in the model as a covariate. It is, in other words, related to potential differences in infectivity between infections that eventually give rise to symptoms and those that remain asymptomatic. Because this statement may continue to be confusing, and is not a core message of our paper, we decided to remove it.

g. The use of 'no significant correlation' is somewhat misleading or too strong since the sample size was (presumably) very small and the problem might well be power to detect such an effect.

The reviewer is correct that our wording may lead to over generalisation. We have adjusted it as follows:

'There was no statistically significant correlation between total parasite density and infectivity (Spearman correlation 0.12, $p=0.081$), reflecting the observed lack of correlation between asexual and gametocyte densities.'

h. Supplemental Fig 1 – many participants did not have feeds and this is not well explained or clear in the main text. Also, why were there no feeds on the days that chronic infections became symptomatic?

The current study was a challenging undertaking where participants had access to care and treatment 24 hours per day and 7 days per week in the study clinic in the field. Mosquito feeding assays were performed in Ouagadougou and participants were transported to the insectary to ensure minimal time between bleeding and mosquito feeding. Nurses accompanied symptomatic individuals to the insectary ensure appropriate clinical monitoring could be provided. Obviously, participant wellbeing was the primary concern and if symptoms occurred outside hours that the insectary was operational, treatment was not delayed. With these challenging logistics, we managed

to conduct 26 feeding assays for the 58 time-points at which individuals were symptomatic. Feeding assays were performed at 19/44 symptomatic episodes for incident infections, and 7/14 for chronic infections. Gametocyte density was assessed for nearly all time-points, including moments of symptomatic infection. This allowed us to examine gametocyte carriage and transmissibility with high confidence.

We have included a statement on the reason why mosquito feeding assays were incomplete for symptomatic infections in the discussion section.

i. It would be a stronger comparison to look at the infectiousness of chronic individuals on days without symptoms and days with symptoms

The reviewer is correct that our current analysis of infectiousness concurrent with symptoms is based on individuals from both cohorts. We have adjusted the relevant section to address the same analysis excluding incident infections.

'...when feeding assays were conducted concurrently with the presentation of symptoms (25/212) infectivity was significantly decreased (risk ratio = 0.30 [95% CI 0.13-0.71], p = 0.0061). Infectivity was significantly lower when symptoms were present even after excluding incident infections (n/N, risk ratio = 0.26 [95% CI 0.11-0.65], p = 0.004).'

3. The authors are trying to make an argument that AP2G copies at day 0 are correlated to gametocytes at d14/rings d0. This seems to be true for chronic infections but not for incident infections. Furthermore, d0 for a chronic infection is not any particular day in the course of their current infection, it just marks the beginning of monitoring so all that can be said from this data is that AP2G copies seem to track with gametocyte density 14 days later in chronic infections. However, this correlation is not rooted in any other context or comparison. Maybe AP2G copies correlate with gametocyte ratio on any day or even on prior days.

In the revised manuscript, we have extended our analysis of transcripts indicative of gametocyte commitment and subsequent appearance of gametocytes. We added another marker, GEXP5, that corroborates findings with AP2G. In addition, we include an analysis where the association between transcript abundance at the first day of intensive follow-up (day 0) and gametocyte production 14 days later is adjusted for gametocyte density at day 0. Associations remain very strong, indicating that transcripts do not reflect the density of mature gametocytes but are indeed indicative of committed rings. We acknowledge that for chronic infections we do not know any of the parasite kinetics prior to intensive follow-up. In addition, as another reviewer pointed out, there are too few data points in incident infections to perform any analysis separated by cohort. We have thus included novel analyses in the results section to address this point. In addition, we acknowledge in the results section that the number of observations in the incident infection cohort was too small to allow an analyses per cohort.

Minor

1. Abstract. Need to mention in the abstract that the observations here may be related to gametocyte maturation.

We have added this hypothesis to abstract as follows:

'We hypothesize that most incident infections were cleared before the density of mature gametocytes was sufficient to infect mosquitoes'

2. Abstract. The last sentence in the abstract is somewhat vague and lacking impact. Yes, we should continue to improve timely case management and screen for infections but we don't need this study to tell us so.

We agree and have removed this sentence.

3. Methods. The ethics statement is confusing. Why are they screening individuals over 12 years if the age range is 5-10 years. Who is providing consent for screening? A 12-year old? Make this clearer

This was an error. The sentence has been amended as follows:

'Individuals whose parent or guardian provided consent for screening were examined by a GCP-trained clinician.'

4. Methods. Be very clear about which measurements were taken with which kind of sample. I can see that finger-stick samples were taken (based on statement later in the methods about extraction from filter paper) but this should be explicit earlier (Sampling section). Later, it mentions qPCR done with whole blood – so was venous blood taken at every visit? In addition, since mosquito feeding surely required venous blood draw, it should be clear which molecular assays are done with venous blood and which are done with finger sticks. Did you use finger-stick and venous blood measurements when calculating PMR?

Re-reading our manuscript, we agree that this was not clearly articulated. With the exception of the days of membrane feeding when venous blood was drawn (day 0, day 14 and day 35) all sampling was by finger prick. All field-based nested PCR assays were performed after extraction from filter paper. All subsequent analyses of parasite density, gametocyte density and gametocyte commitment markers were done on nucleic acids extracted from whole blood collected in microtainers by finger pricks. This is clarified in the revised manuscript.

5. Results. Use enrollment and recruitment consistently. I'm confused in Table 1 where recruitment density is same between chronic and incident cohort. I think this means density at day 0 for chronic but density at day of treatment for incident? Or does it mean density at day 0 for both? Is day 0 recruitment or enrolment or neither?

We acknowledge that these terms were used interchangeably, and the time-point they referred to was not always clear in the first manuscript. We have changed figure 1 and all language referring to enrolment. The figure and text now make clear that day 0 of the intensive follow-up = enrolment, and all preceding sampling was conducted as screening only.

6. Results. Indicate how density was measured in Table 1. I assume by qPCR

Density was measured indeed measured by qPCR or by qRT-PCR. This is now indicated in the table.

'Asexual parasite density was measured by 18S ribosomal DNA based qPCR [55]. Symptoms were defined as fever (>37.5°C), or other clinical manifestations of malaria infection. Clonal diversity was determined by MSP2 genotyping⁶¹. Gametocyte densities were measured by quantification of female (Pfs25) and male (PfMGET) gametocyte specific mRNA by qRT-PCR⁵⁶.'

7. Results. Following the comment above, were these individuals similar at screening – ie parasite density by qPCR? In other words, were these cohorts essentially the same at screening and then only differentiated by presumptive tx? How many of the chronic cohort had microscopically-detectable parasites?

Full details of parasitaemia at enrolment are provided in Table 1.

8. Results. Following the comment above, it would have been interesting to see what would have

happened if the chronic cohort, after completing the follow-up, was treated and then followed for incident infection. Are these groups essentially identical and the differences in PMR, gametocytes and infectivity related to acquisition of a new infection or perhaps just the age of the infection/maturation of gametocytes?

This is an interesting and valuable suggestion. It would have required a different study design and longer time of follow-up to maximize the chance that participants whose chronic infections were treated had a chance of acquiring an incident infection following the end of the prophylactic period provided by first line treatment. This was beyond the scope of the current manuscript.

9. Results. The red line in Figure 2B that shoots up doesn't have a black dot but stops at about 10-12 days. Who is this? Not one of the incident cohort that had no symptoms by Day 35. Is the dot missing or was this person lost to follow-up? I guess this is the same participant as the lower left panel in supplemental figure 2A?

There was an error in the presentation of data for 3 individuals with clinical infection, as described in response to a point made by reviewer 1 about supplemental figure 2 (wherein, time points with clinical infection did not appear in figures because parasite density measures were absent at these time-points – due to sample loss). In figure s2, we are able to address this by including the symptomatic markers as asterisks at the appropriate time-points, and providing full details of these in the legend. For figure 2, as all individuals are present in the same graph, we have addressed the absence of the spots in the legend but have not altered the graph; adding a symptomatic time-point on a specific line of a density based plot would not accurately reflect the available data.

10. Discussion. The statement that asymptomatic individuals contribute more to the reservoir is too strong given the very limited age range in this study.

This is a valid point. Whilst the current findings are in line with our previous work on this topic, all pointing towards a smaller contribution of symptomatic infections to transmission, the reviewer is correct that this conclusion is not warranted based on our findings here. The current study did not quantify the human infectious reservoir for malaria but studied in detail two important aspects of this reservoir: gametocytaemia and infectivity in relation to time since infection and symptoms.

11. Supplemental Fig 1 – shouldn't everyone in B have a red dot?

The reviewer is correct, and in fact everyone in B does have a red dot, some of them were just less clear because they are under 'mosquito feeding assay' crosses. We have edited these graphs to ensure red dots are all visible.

12. Supplemental Fig 2. Why do some participant's trajectories stop before day 35 with no red dot (supplemental fig 2B)

Correct – this was overlooked. We have addressed this point in response to reviewer 1 and in response to point 9 above. Supplemental figure 2 has been adjusted.

Very minor/grammar or style edits

- 1. First two sentences of the intro start with describing something as “poorly understood**
- 2. Methods, Data Analysis: changed should be changes**

These points have been addressed.

Reviewer #3 (Remarks to the Author):

This is a fascinating report of an intensive surveillance study of the dynamics of infection and infectiousness in a cohort of school-aged children in Burkina Faso.

The scientific methods are sound and rigorous. However, I struggle to understand the specific questions and/or hypotheses being addressed and wonder if the findings of this study are generalizable.

My specific major concerns are:

1. Incident and chronic infections were treated differently. Children with incident infections were treated with long-acting dihydroartemisinin-piperaquine which may have impacted the subsequent infection dynamics and infectiousness (especially as the median time to infection was relatively brief).

The reviewer is correct in stating that children with incident infection received presumptive DP treatment at the first screening visit. The median time from screening to enrollment was 27 days (IQR 20-41). The duration of protection at a level of 90% or higher is ~ 16.8 days for piperaquine whilst the duration of protection at 50% or over is ~ 25.8 days for piperaquine (Okell et al. DOI: 10.1038/ncomms6606). There is strong evidence from controlled infection models that piperaquine does not affect developing or mature gametocytes (Alkema et al. doi: 10.1093/infdis/jiaa157; Reuling et al. doi: 10.7554/eLife.31549; Collins et al. doi: 10.1172/JCI98012). Whilst low levels of piperaquine will thus not have affected gametocyte development, it cannot be ruled out that incident infections developed under very low levels of piperaquine in some participants. If so, parasite multiplication rates in the incident cohort, already considerably higher than in chronic infections, might have been even higher. This uncertainty is mentioned in the revised discussion section.

'Most incident infections occurred within 2 months (interquartile range 20-41 days) after presumptive treatment with dihydroartemisinin-piperaquine, when residual piperaquine levels are unlikely to provide effective prophylaxis [38]; piperaquine has no known effect on (developing) gametocytes [27].'

2. It is difficult to disentangle the host vs. parasites vs. host-parasite interaction factors that contributed to children becoming incident vs. chronic infections. Thus, the studies many parameters (antibody level, PMR, ap2-g level) highlight associations but offer no insight into causation.

The reviewer highlights a challenge with all observational studies. It is indeed difficult to disentangle all factors. However, our findings of lower gametocyte commitment transcripts in incident infections, lower gametocyte density in incident infections and lower gametocyte infectivity when mosquitoes are feeding at the moment of symptoms (more common in incident infections) all point towards lower transmission potential in incident infections. In our revised manuscript, we pay extra attention to avoid any further conclusions on causality or mechanisms and make it clear where we interpret our data or speculate on possible (not proven) mechanisms.

3. I may have misunderstood, but the timing of the mosquito infectivity assay was not standardized across cohorts.

The timing was standard, but our language may have been unclear. We have amended the results section to state:

'Blood samples were taken from individuals with confirmed incident or chronic infection at enrolment (day 0), then daily for 1 week, and weekly until day 35 or until the presentation of malaria symptoms

(figure 1). Infectivity to mosquitoes was determined at day 0, 14 and 35, or at the final visit if this preceded day 35 due to the presentation of symptoms.'

4. The figures provide details on every measurement but it remains unclear what the conclusions are with so many data points.

In response to other reviewer comments, we have clarified some of our figures and added (supplemental) tables to provide details of individual infections (whilst focusing on general trends in our interpretation) and more easily digestible summary measures per cohort.

5. The manuscript does not explain why these specific outcomes were selected, as the introduction seems to focus on symptomatic status and gametocyte dynamics.

It is evident from this comment, as well as from other comments of reviewers, that our hypotheses and choice of outcome measures were not clearly described in the original submission. We have re-written large parts of the introduction and results section to improve clarity and are confident that this has resulted in a clearer manuscript.

I have identified several specific areas that could be addressed:

Abstract

- **Would be helpful to define incident vs. chronic cohort**
- **Define prospective gametocyte production**

These are now defined clearly in the introduction and methods. Figure 1 has been simplified and should assist in distinguishing the cohorts.

Results

Incident infections occurred with a median of ~4 weeks. What occurred during the follow up time after the incident infection?

The intensive follow up period for incident infections started at the point when incident infections were observed (i.e. all study data comes from this period of follow up, not the screening period beforehand). We hope this is now more clear in the adjusted version of figure 1.

Figure 1 is very detailed and difficult to follow. A more simplified depiction of the study design might help the reader to understand.

We agree this figure was difficult to follow, and have simplified it.

There was no mention of the relationship between gametocyte sex ratio and infectiousness. This could be an interesting piece of data.

We left this out of the original manuscript because we have analysed the (complex) relationship between sex ratio and infectivity with a large sample size in great detail elsewhere (<https://elifesciences.org/articles/34463>). We have now provided a simple analysis and show that there is no direct relationship between sex ratio and infectivity among this study population, as shown below. It would have been very interesting to see if this effect was similar between cohorts, but unfortunately there are too few infections in the incident cohort to allow for this comparison.

'There was no evidence that optimal sex ratios (between 10-50% male) were associated with higher mosquito infection rates ($p=0.634$); the relationship between sex-ratio and infectivity is complex, and has been explored in greater detail elsewhere²⁴.'

Table 2 presents coefficients that are difficult to interpret. What was the baseline distribution of hemoglobin type?

Prevalence of haemoglobin types for all participants has been added to Table 1.

Mosquito infectivity assays: A series of bivariate analyses are presented (chronic vs incident, symptomatic vs asymptomatic), but it is not clear if these contribute independently or there is confounding.

All analyses of infectivity rates were adjusted for asymptomatic vs symptomatic and chronic vs incident ('cohort'). These effects are therefore indeed independent. We have added the section in bold in our results section to highlight this.

'Gametocyte densities were generally higher in chronic infections, but after adjustment for gametocyte density the risk of mosquito infection for individuals in the chronic cohort remained higher (risk ratio = 10.44 [95% CI 2.79-39.01], p = 0.0005); this was true even after further adjusting for the presence of symptoms (risk ratio = 11.17 [95% CI 3.08-40.48], p = 0.0002).'

Methods

Please clarify the reason for differentiating qPCR and nPCR?

We consider the rapid turn-around of molecular diagnosis of malaria a major strength of our study, allowing us to capture infections early. For this, nPCR was done on site. At the moment we conducted the study, qPCR was not possible on site. To quantify parasite burden and confirm nPCR results, we later performed qPCR on freshly extracted nucleic acids. We have clarified our methodology in the revised manuscript as follows:

'The use of nPCR during screening and for infection confirmation was important for rapid sample turn-around. Higher sensitivity qPCR assays (also 18s based) were performed later at RadboudUMC, and the results of these assays supercede those of nPCR for all analyses.'

Mosquito feeding assays were performed without serum replacement. Were sexual stage serological responses assessed? Only erythrocytic antigens were presented though a major focus on the manuscript was gametocyte production and infectiousness.

As part of our revision, we have extended our serological analyses to also include key sexual stage antigens (Pfs48/45 and Pfs230). Antibody responses to these antigens had a profound effect on transmission potential, thus adding valuable new components to our revised manuscript: Infectivity was 65% (95% CI 21-85) lower for a 10 fold increase in anti-Pfs48/45 antibody concentration (p=0.011); for Pfs230, infectivity was 64% (95% CI 34-80) lower for a 10 fold increase. These findings now receive prominence in the revised results section.

Reviewer comments,second round –

Reviewer #1 (Remarks to the Author):

The authors have addressed my original queries.

Reviewer #3 (Remarks to the Author):

The revised manuscript adequately addresses the concerns that the other reviewers and I raised. I have not further comments.